JCB Journal of Cell Biology

# High-speed single-molecule imaging reveals signal transduction by induced transbilayer raft phases

Ikuko Koyama-Honda[1], Takahiro K. Fujiwara[2], Rinshi S. Kasai[3], Kenichi G.N. Suzuki[2,4,5], Eriko Kajikawa[6], Hisae Tsuboi[7], Taka A. Tsunoyama[7], and Akihiro Kusumi[7]

Using single-molecule imaging with enhanced time resolutions down to 5 ms, we found that CD59 cluster rafts and GM1 cluster rafts were stably induced in the outer leaflet of the plasma membrane (PM), which triggered the activation of Lyn, H-Ras, and ERK and continually recruited Lyn and H-Ras right beneath them in the inner leaflet with dwell lifetimes <0.1 s. The detection was possible due to the enhanced time resolutions employed here. The recruitment depended on the PM cholesterol and saturated alkyl chains of Lyn and H-Ras, whereas it was blocked by the nonraftophilic transmembrane protein moiety and unsaturated alkyl chains linked to the inner-leaflet molecules. Because GM1 cluster rafts recruited Lyn and H-Ras as efficiently as CD59 cluster rafts, and because the protein moieties of Lyn and H-Ras were not required for the recruitment, we conclude that the transbilayer raft phases induced by the outer-leaflet stabilized rafts recruit lipid-anchored signaling molecules by lateral raft–lipid interactions and thus serve as a key signal transduction platform.

## Introduction

In the human genome, >150 protein species have been identified as glycosylphosphatidylinositol (GPI)-anchored proteins, in which the protein moieties located at the extracellular surface of the plasma membrane (PM) are anchored to the PM by way of GPI, a phospholipid (Kinoshita and Fujita, 2016). Many GPI-anchored proteins are receptors and thus are referred to as GPI-anchored receptors (GPI-ARs). A GPI-anchored structure appears paradoxical for receptors because it spans only halfway through the membrane; yet, to function as a receptor, it has to relay the signal from the outside environment to the inside of the cell (Fig. 1 A). "Raft domains" are PM domains on the space scales from a few nanometers up to several hundred nanometers that are built by cooperative interactions of cholesterol and molecules with saturated alkyl chains of C16 or longer, as well as by their exclusion from the bulk unsaturated chain–enriched domains (Kusumi et al., 2020; Levental et al., 2020), have been implied in the signaling process of GPI-ARs across the PM (Omidvar et al., 2006; Suzuki et al., 2007b, 2012; Paulick and Bertozzi, 2008; Eisenberg et al., 2011; Fessler and Parks, 2011; Lingwood et al., 2011; Kusumi et al., 2014; Raghupathy et al., 2015). Nevertheless, exactly how raft domains or raft-based lipid interactions participate in the transbilayer signal transduction of GPI-ARs remains unknown. Indeed, raft-based

interactions might even be involved in the signal transduction by transmembrane (TM) receptors (Coskun et al., 2011; Chung et al., 2016; Shelby et al., 2016).

In giant unilamellar vesicles undergoing liquid-ordered (Lo)/liquid-disordered (Ld) phase separation, the Lo/Ld phase domains in the outer leaflet spatially match the same domains in the inner leaflet, indicating strong interbilayer coupling due to phase separation across the bilayer (Collins and Keller, 2008; Blosser et al., 2015). In living cells, the long-chain phosphatidylserine present in the PM inner leaflet was proposed to play key roles in the transbilayer coupling (Raghupathy et al., 2015). However, the mechanisms of transbilayer coupling in the PM for the induction of signal transduction are not well understood.

Using CD59 as a prototypical GPI-AR, our previous single–fluorescent molecule imaging showed that nanoparticle-induced CD59 clusters form stabilized raft domains with diameters on the order of 10 nm in the PM outer leaflet, which in turn continually recruit intracellular signaling molecules Giα, Lyn, and PLCγ2 one after another in a manner dependent on raft–lipid interactions, triggering the inositol triphosphate/Ca²⁺ signaling pathway. Namely, artificially induced CD59 clusters behaved like CD59 clusters induced by the addition of the complement component C8 or the membrane attack complement complexes

[1]Department of Biochemistry and Molecular Biology, Graduate School and Faculty of Medicine, University of Tokyo, Tokyo, Japan; [2]Institute for Integrated Cell-Material Sciences, Kyoto University, Kyoto, Japan; [3]Institute for Frontier Life and Medical Sciences, Kyoto University, Kyoto, Japan; [4]Institute for Glyco-core Research, Gifu University, Nagoya, Japan; [5]Center for Highly Advanced Integration of Nano and Life Sciences, Gifu University, Gifu, Japan; [6]Laboratory for Organismal Patterning, Center for Biosystems Dynamics Research, RIKEN Kobe, Kobe, Japan; [7]Membrane Cooperativity Unit, Okinawa Institute of Science and Technology Graduate University, Onna-son, Okinawa, Japan.

Correspondence to Akihiro Kusumi: akihiro.kusumi@oist.jp.



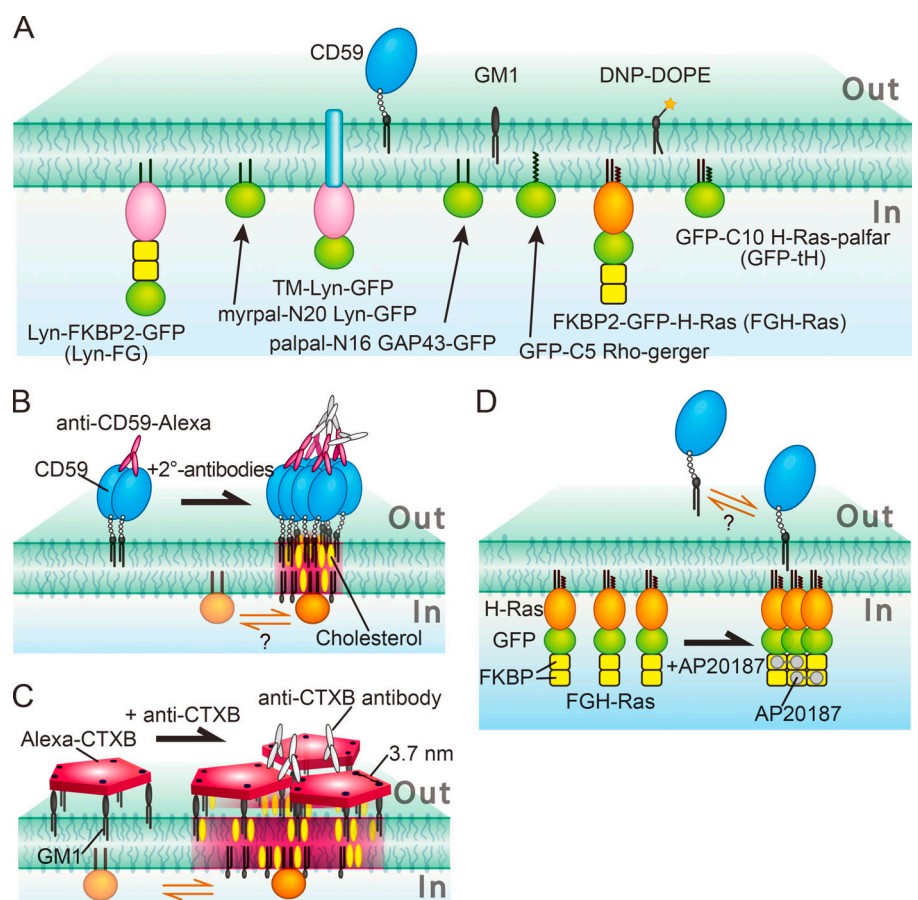

Figure 1. **Outer- and inner-leaflet lipid-anchored molecules employed in this study and their cross-linking schemes. (A)** The outer-leaflet molecules employed in this work were a prototypical GPI-AR, CD59; a prototypical ganglioside, GM1; and a prototypical nonraft phospholipid, DNP-DOPE. The inner-leaflet molecules examined here were (G and GFP represent EGFP) the following: Lyn-FG, Lyn conjugated at its C-terminus to two molecules of FKBP in series and then to GFP; Myrpal-N20Lyn-GFP, myristoyl, palmitoyl-anchored Lyn peptide conjugated to GFP, where the peptide was the 20-aa N-terminal sequence of Lyn, which contains the conjugation sites for both myristoyl and palmitoyl chains; TM-Lyn-GFP, the TM mutant of Lyn-GFP, in which the TM domain of a prototypical nonraft molecule LDLR was conjugated to the N-terminus of the full-length Lyn-GFP (which cannot be fatty acylated); Palpal-N16GAP43-GFP, palmitoyl, palmitoyl-anchored GAP43 peptide conjugated to GFP, in which the peptide was the 16-aa N-terminal sequence of GAP43 containing two palmitoylation sites (likely to be raft associated); GFP-C5Rho-geranylgeranyl, GFP anchored by a geranylgeranyl chain, in which GFP was conjugated at its C-terminus to the five-aa C-terminal sequence of Rho, which contains a site for attaching an unsaturated geranylgeranyl chain (likely to be non–raft associated); FGH-Ras, H-Ras chimera molecule in which two tandem FKBP molecules linked to GFP were then conjugated to H-Ras; and GFP-tH, GFP linked to the 10-aa C-terminal sequence of H-Ras containing two sites for palmitoylation and a site for farnesylation. These molecules were expressed and observed in live HeLa cells. **(B–D)** The schemes for clustering (cross-linking) CD59 (B), GM1 (C), and FGH-Ras (D). CD59 was clustered by the sequential additions of anti-CD59 mAb IgG labeled with the fluorescent dye A633 and secondary Abs (+2°-antibodies; B). GM1 was clustered by the sequential additions of CTXB conjugated with A633 and anti-CTXB Abs (C). FGH-Ras (as well as Lyn-FG) was clustered by the addition of AP20187 (cross-linker for FKBP; D). After the induction of clustering of these molecules, the possible recruitment of lipid-anchored molecules in the other leaflet of the PM at these clusters was examined.

(MACCs; Suzuki et al., 2007a, 2007b, 2012). Therefore, the CD59 clusters were termed "CD59 cluster signaling rafts" or simply "CD59 cluster rafts" (Stefanová et al., 1991; Suzuki et al., 2007a, 2007b, 2012; Simons and Gerl, 2010; Zurzolo and Simons, 2016). Importantly, the recruitment of cytoplasmic signaling molecules at the CD59 signaling rafts occurred transiently, in a time scale on the order of fractions of a second (in the following text, we use the expression "recruitment of signaling molecules 'at' CD59 clusters" rather than "the recruitment 'to' CD59 clusters" because our imaging method could not directly show the binding of the signaling molecules located in the inner leaflet to the CD59 clusters located in the outer leaflet). Raftlike properties of the artificial antibody (Ab)-induced CD59 clusters were confirmed by the finding that fluorescently labeled gangliosides and sphingomyelins are colocalized with the artificial CD59 clusters (Komura et al., 2016; Kinoshita et al., 2017). CD59-TM, in which the GPI anchor was replaced by the TM domain of a prototypical nonraft molecule, low-density lipoprotein receptor (LDLR), failed to exhibit the raftlike behaviors and to trigger the downstream signal, in ways similar to the CD59 clusters after cholesterol depletion (Suzuki et al., 2007a, 2007b, 2012). The

present research was designed on the basis of these previous research results. Furthermore, our previous single-molecule studies revealed that, although gangliosides and sphingomyelins are always present in the CD59 cluster signaling rafts, each lipid molecule associates with the CD59 cluster raft for only 50–100 ms (Komura et al., 2016; Kinoshita et al., 2017), like signaling molecules Giα, Lyn, and PLCγ2.

Meanwhile, the time resolution of the single-molecule imaging method used to detect such transient colocalization events was only 33 ms. In the present study, we greatly enhanced the imaging time resolutions down to 5.0 and 6.45 ms, an improvement by factors of 6.7 and 5.2, respectively, and thus substantially refined the detection of cytoplasmic signaling molecule colocalizations with CD59 cluster rafts (and GM1 cluster rafts). To the best of our knowledge, these are likely to be the fastest simultaneous, two-color, single-molecule observations ever performed. We previously found Lyn recruitment at CD59 cluster rafts, but in the present research, by applying single-molecule imaging at enhanced time resolutions and using various lipid-anchored cytoplasmic molecules, including Lyn, H-Ras, and four artificially designed molecules, as

well as by using the stabilized ganglioside GM1 cluster rafts in addition to the CD59 cluster rafts, we sought to unravel the mechanisms by which cytoplasmic lipid-anchored signaling molecules in the PM inner leaflet are recruited at CD59 cluster rafts and GM1 cluster rafts formed in the PM outer leaflet.

In addition to the well-known function of CD59 to protect normal cells in the body against self-attack by MACCs, CD59 is involved in tumor growth. First, CD59 renders autologous carcinoma cells insensitive to the MACC action, providing tumor cells with a key strategy to evade the immune system (Morgan et al., 1998; Carter and Lieber, 2014). Second, the MACC-induced CD59 clusters activate the extracellular signal-regulated kinase (ERK) signaling pathway, thus enhancing tumor cell proliferation (Jurianz et al., 1999). Therefore, the basic understanding of CD59 signaling, particularly the Lyn (Src family kinase) signaling to trigger the inositol triphosphate/Ca²⁺ pathway for protection against MACC binding, as well as the signaling cascades for ERK activation by way of Lyn and Ras (Bertotti et al., 2006; Harita et al., 2008; Wang et al., 2011; Suzuki et al., 2012; Croucher et al., 2013; Dorard et al., 2017), would be useful for developing methods to regulate CD59 function, eventually leading to better therapeutic outcomes in oncology by suppressing ERK activities and reversing complement resistance (Carter and Lieber, 2014).

In the present research, we first aimed to unravel how the CD59 cluster rafts in the PM outer leaflet recruit the downstream intracellular lipid-anchored signaling molecules Lyn and H-Ras, located in the PM inner leaflet. Lyn is anchored to the PM inner leaflet by a myristoyl chain and a palmitoyl chain (myrpal), whereas H-Ras is anchored by two palmitoyl chains and a farnesyl chain (Fig. 1 A). Because CD59 cannot directly interact with and activate Lyn and H-Ras, and because Lyn and H-Ras are proposed to be raft domain associated in the PM inner leaflet (Field et al., 1997; Sheets et al., 1999; Prior et al., 2001, 2003), we paid special attention to raft–lipid interactions as a recruiting mechanism (Wang et al., 2005) while also considering protein–protein interactions (Fig. 1 B; Douglass and Vale, 2005).

Second, to directly examine the possibility that the signal transfer from the PM outer leaflet to the inner leaflet is mediated by raft–lipid interactions, we cross-linked the prototypical raft lipid ganglioside GM1 in the outer leaflet to examine whether GM1 clusters could recruit Lyn and H-Ras in the inner leaflet (Fig. 1, A and C). Many studies have examined the cytoplasmic signals triggered by GPI-AR stimulation and GM1 clustering in a raft-dependent manner (Pyenta et al., 2001; McKerracher and Winton, 2002; Wang et al., 2005; Todeschini et al., 2008; Fujita et al., 2009; Um and Ko, 2017), although the results varied considerably. In contrast, very few studies have investigated the actual recruitment of cytoplasmic lipid-anchored signaling molecules at the stabilized nanoraft domains formed in the PM outer leaflet (Harder et al., 1998; Suzuki et al., 2007a, 2007b, 2012), and particularly the molecular dynamics of the recruitment in live cells. In the present study, as a control, we induced the clustering of lipid-anchored Lyn or H-Ras in the PM inner leaflet and observed whether this could induce the recruitment of CD59 and GM1 in the PM outer leaflet (Fig. 1, A and D).

## Results

### Ab-induced CD59 clusters in the PM and their ERK activation

First, we improved the time resolution of our home-built single-molecule imaging station, described previously (Koyama-Honda et al., 2005; Komura et al., 2016; Kinoshita et al., 2017). The improvements were accomplished by using two kinds of camera systems that can operate at higher frame rates (see Materials and methods) and modifying the single-molecule imaging station by using lasers with higher outputs and tuning the excitation optics. As a result, the time resolution was enhanced from 33.3 ms (30 Hz) to 5.0 or 6.45 ms (200 or 155 Hz, respectively, which is faster than normal video rate by factors of 6.7 and 5.2, respectively), with frame sizes of 640 × 160 pixels and 653 × 75 pixels, respectively. We employed the same two cameras for performing simultaneous, two-color, single-molecule imaging (see Materials and methods). Throughout this work, all of the microscopic observations of CD59 cluster rafts (Alexa Fluor 633 [A633] tagged) and the downstream cytoplasmic signaling molecules (fused to EGFP, which is simply called "GFP" for conciseness) were performed simultaneously in the bottom (basal) PM of HeLa cells.

CD59 cluster signaling rafts were formed by the addition of the primary (anti-CD59 IgG mAb conjugated with A633) and secondary Abs, according to previous reports (Field et al., 1997; Janes et al., 1999; Chen and Williams, 2013). Using this method, CD59 clusters could be formed in both the apical and basal PMs, whereas in our previous method of using nanoparticles to induce CD59 clusters, due to the nonaccessibility of the particles in the space between the basal PM and the coverslip, CD59 clusters were formed only in the apical PM. Therefore, in this study, we observed the CD59 clusters and signaling molecules in the basal PM, which enabled observations with improved signal-to-noise ratios. These observations were conducted within 10 min after the addition of the secondary Abs, when more than 92% of the CD59 clusters were located outside caveolae (Fig. S1 A).

To better observe the short-term colocalizations of lipid-anchored signaling molecules with CD59 cluster rafts, we hoped to slow down the colocalization processes, and therefore all microscopic observations were performed at 27°C, which is 10°C lower than the physiological temperature of 37°C. It is known that raft formation is temperature dependent, but in all the cell lines examined thus far, the temperature-dependent changes are pronounced below ~15°C, at which large Lo phase–like raft domains are induced and become visible by fluorescence microscopy (for visualization, actin-based membrane skeleton meshes must be removed from the PM cytoplasmic surface); this would not occur at 27°C (Holowka and Baird, 1983; Gidwani et al., 2001; Veatch and Keller, 2003; Baumgart et al., 2007; Lingwood et al., 2008; Sengupta et al., 2008; Levental et al., 2009; Kusumi et al., 2020). Namely, the changes found in the PM when the temperature is lowered from 37°C to 27°C would be quantitative rather than qualitative. For example, the diffusion coefficients of various lipids and GPI-ARs in two very different cell types, CHO and rat basophilic leukemia (RBL)-2H3 cells, were reported to decrease only by a factor of at most 1.4 when the temperature was lowered from 37°C to 27°C (Lee et al., 2015; Saha et al., 2015). Meanwhile, the diffusion coefficients of both the prototypical

nonraft phospholipid L-α-dioleoylphosphatidylcholine (DOPE) and the prototypical raft-associated phospholipids C18-sphingomyelin and L-α-distearoylphosphatidylcholine (all of them fluorescently labeled) would be reduced by a factor of approximately 2 when the temperature was lowered from 37°C to 27°C (assuming that the activation energy for diffusion is the same between 37°C and 23°C; Kinoshita et al., 2017, where we used T24 and PtK2 cells; on the basis of these results, we decided to perform all of the microscopic observations at 27°C to better detect the colocalization processes). Therefore, we believe that the conclusions obtained in the present work based on the observations performed at 27°C are essentially correct.

The number of CD59 molecules located in a CD59 cluster was estimated to be ~10 (molecules) on average (the variations would be quite large; Fig. 2, A and B; Materials and methods). Because CD59 is anchored to the PM outer leaflet by way of two saturated, long alkyl chains, the CD59 clusters employed here would contain an average of 20 saturated long alkyl chains of CD59 in the small cross-sectional area of the CD59 cluster. The CD59 clusters diffused at a threefold slower rate than monomeric CD59 (labeled with anti-CF59–antigen-binding fragment [Fab]-A633; Fig. 2, C and D). Because we used the dye (A633)-conjugated Ab (and the secondary Abs) to induce CD59 clusters, the recording periods were quite limited due to photobleaching (~0.51 s), and signal-to-noise ratios for observing the CD59 clusters were worse than with our previous observations using fluorescent nanoparticles. In the present study, we could not detect stimulation-induced temporary arrest of lateral diffusion, and the CD59 clusters appeared to simply undergo slow diffusion.

CD59 clustering triggered the signaling cascade to activate the ERK1/2 kinases (performed at 37°C instead of 27°C; Fig. 3), in agreement with a previous finding (Jurianz et al., 1999). The signaling pathways leading to ERK activation could involve the small G-protein H-Ras, as well as Lyn (Bertotti et al., 2006; Harita et al., 2008; Porat-Shliom et al., 2008; Wang et al., 2011; Croucher et al., 2013; Dorard et al., 2017). Therefore, we performed direct single-molecule observations of the recruitment of both Lyn kinase and H-Ras to the CD59 cluster signaling rafts. We had previously detected Lyn recruitment at CD59 clusters (Suzuki et al., 2007a, 2007b), but in the present research we focused on understanding the recruitment mechanism by using other related molecules and H-Ras, as well as by using single-molecule observations with improved time resolutions.

### Lyn is continually and transiently recruited at CD59 cluster rafts one molecule after another, but not at nonclustered CD59

Lyn is anchored to the PM inner leaflet by myristoyl and palmitoyl chains conjugated to its N-terminus (Fig. 1 A). The Lyn conjugated at its C-terminus to two molecules of FK506-binding protein (FKBP) in series and then to GFP (Lyn-FG) used here for single-molecule observations would be functional because it could be phosphorylated in RBL-2H3 cells after antigen (DNP) stimulation (Fig. S2 A). Virtually all of the Lyn-FG molecules on the PM inner leaflet were monomers (undergoing a single-step photobleaching like GFP molecules sparsely adsorbed on the

glass; Fig. S3) and underwent thermal diffusion, with a mean diffusion coefficient (in the time scale of 124 ms) of 0.76 ± 0.0019 $\mu m^2/s$ (Fig. 4 A).

Simultaneous two-color single-molecule observations revealed that Lyn-FG molecules diffusing in the inner leaflet were continually recruited at CD59 clusters located in the PM outer leaflet, one molecule after another. Importantly, the dwell time of each Lyn-FG molecule at the CD59 cluster was on the order of 0.1 s (Fig. 5 and Video 1). Quantitative detection of colocalizations was performed by using our previously developed definition, in which fluorescent spots with two different colors are located within 150 nm (Koyama-Honda et al., 2005). Although the colocalization distance of 150 nm is clearly much greater than the sizes of the interacting molecules, which would generally be on the order of several nanometers, the colocalization analysis is still useful for detecting molecular interactions for the following reason. Unassociated molecules may track together by chance over short periods of time for short distances, but the probability of this occurring for multiple frames is small. Therefore, longer colocalization durations imply the presence of molecular interactions between the two molecules rather than incidental encounters (although molecular interactions are initiated by incidental encounters; see Materials and methods).

Each time we detected a colocalization event of an Lyn-FG molecule with a CD59 cluster, we measured its duration, and after observing sufficient numbers of colocalization events, we obtained a histogram showing the distribution of colocalized durations for Lyn-FG and CD59 clusters (Fig. 6 A, a; Materials and methods). However, this duration histogram must also contain the colocalization events due to incidental close encounters of molecules within 150 nm, without any molecular interactions. To obtain the histogram of incidental colocalization durations, the image obtained in the longer-wavelength channel (A633) was shifted toward the right by 20 pixels (1.0 and 1.19 $\mu$m, depending on the camera) and then overlaid on the image obtained in the GFP channel ("shifted overlay"). The duration histogram for incidental colocalization, called $h$(incidental-by-shift), could effectively be fitted with a single exponential function with a decay time constant $\tau_1$ of 15 ± 0.93 ms (Throughout this report, the SEM of the dwell lifetime is provided by the fitting error of the 68.3% confidence limit for the decay time constant).

The distribution of the durations obtained for correctly overlaying the Lyn-FG movies and CD59 cluster movies was significantly different from that for the shifted overlay (P = 0.00076 using the Brunner-Munzel test; Brunner and Munzel, 2000; throughout this report, the Brunner-Munzel test was used for the statistical analysis, and all statistical parameters are summarized in Table S1, Table S2, and Table S3). The histogram of colocalization durations for Lyn-FG at CD59 clusters could be fitted with the sum of two exponential functions with decay time constants of $\tau_1$ and $\tau_2$. In the fitting, $\tau_1$ was preset as the decay time constant determined from $h$(incidental-by-shift), and $\tau_2$ was determined as a free parameter. In the previous studies using normal video rate (30 Hz; 33-ms resolution), due to insufficient time resolutions, such distinct components could not be observed in the colocalization duration histogram.

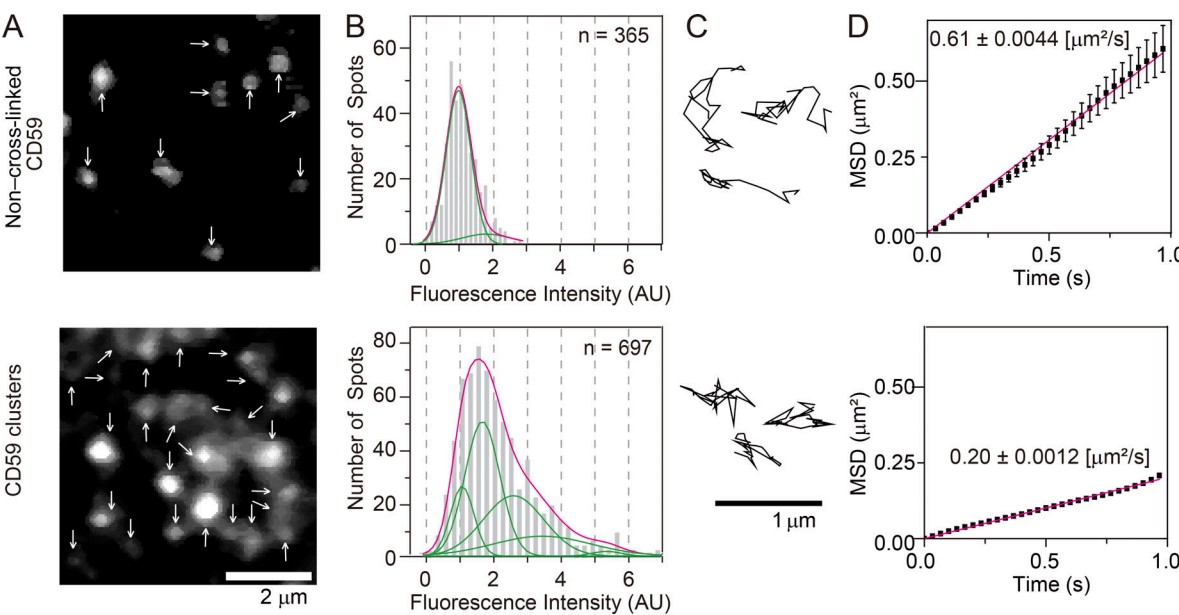

Figure 2. **CD59 clusters in the PM outer leaflet contained an average of ~10 CD59 molecules and diffused slowly. (A)** Fluorescence images of non–cross-linked CD59 bound by A633–anti-CD59 Fab (D/P, 0.27; top) and CD59 clusters induced by the sequential additions of A633–anti-CD59 IgG (D/P, 0.63) and the secondary Abs (bottom), obtained at single-molecule sensitivities. Arrows indicate all of the detected fluorescence spots in each image. **(B)** Histograms showing the distributions of the signal intensities of individual fluorescence spots of non–cross-linked CD59 (Fab-A633 probe; top, $n$ = 355) and CD59 clusters (bottom, $n$ = 697). On the basis of these histograms, we concluded that each CD59 cluster contained an average of ~10 CD59 molecules (see Materials and methods), although the number distributions would be quite broad. **(C)** Typical trajectories of non–cross-linked CD59 (top) and CD59 clusters (bottom) for 0.2 s, obtained at a time resolution of 6.45 ms. **(D)** Ensemble-averaged mean-square displacements (MSDs) plotted against time, suggesting that in the time scale of 1 s, both non–cross-linked and clustered CD59 (68 and 119 trajectories, respectively) undergo effective simple Brownian diffusion, and the diffusion is slowed by a factor of about 3 after Ab-induced clustering. All error bars represent SEM.

As described in Materials and methods, $\tau_2$ directly represents the binding duration (inverse off-rate assuming simple zero-order dissociation kinetics for Lyn-FG from the CD59 cluster). Although the errors involved in the determinations of $\tau_2$ are

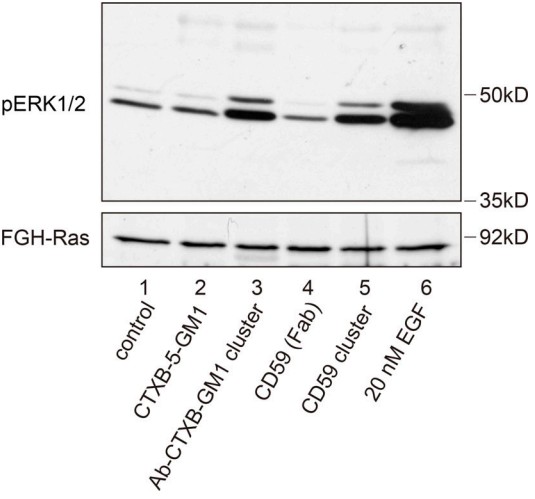

Figure 3. **Both CD59 clusters and GM1 clusters induced by the sequential additions of CTXB and its polyclonal Abs (Ab-CTXB-GM1 clusters) induced Erk phosphorylation (activation).** Note that the simple clustering of five GM1 molecules by CTXB (CTXB-5-GM1) failed to trigger ERK activation. Western blotting was performed by using antiphosphorylated Erk Abs (top) with anti-H-Ras Abs as the loading controls (bottom). The addition of 20 nM EGF was used as a positive control for Erk activation.

quite large due to the problem of signal-to-noise ratios of the images, we emphasize that, within the scope of this report, the presence or absence of the $\tau_2$ components in the colocalization duration histograms would already be of key importance.

In the case of the colocalizations of Lyn-FG with CD59 cluster rafts, the fitting provided a $\tau_2$ of 80 ± 25 ms (Table S1). Both $\tau_1$ and $\tau_2$ for all of the molecules investigated here were much shorter than the photobleaching lifetimes of GFP and A633 (>400 ms; i.e., 62 or 80 image frames), and therefore no corrections for photobleaching were performed in this research. The 80-ms dwell lifetime of Lyn at CD59 clusters is shorter than that observed previously (median, 200 ms; Suzuki et al., 2007a), probably due to the improved time resolutions and signal-to-noise ratios (previously, shorter colocalizations were likely missed) as well as the different ways of forming CD59 clusters. Therefore, this result indicates that Lyn is recruited at CD59 cluster rafts more transiently than we previously evaluated.

Next, the colocalizations of Lyn-FG with nonclustered CD59 (labeled with anti-CD59 Fab-A633) were examined. The duration histogram obtained by the correct overlay was almost the same as $h$(incidental-by-shift) (P = 0.86; $\tau_1$ = 19 ms; Fig. 6 A, b, and Table S1), and it was significantly different from the histogram with CD59 clusters (P = 0.018).

**Lyn recruitment at CD59 clusters requires raft–lipid interactions**

Next, we asked whether raft–lipid interactions and protein–protein interactions are required for recruiting Lyn-FG at CD59

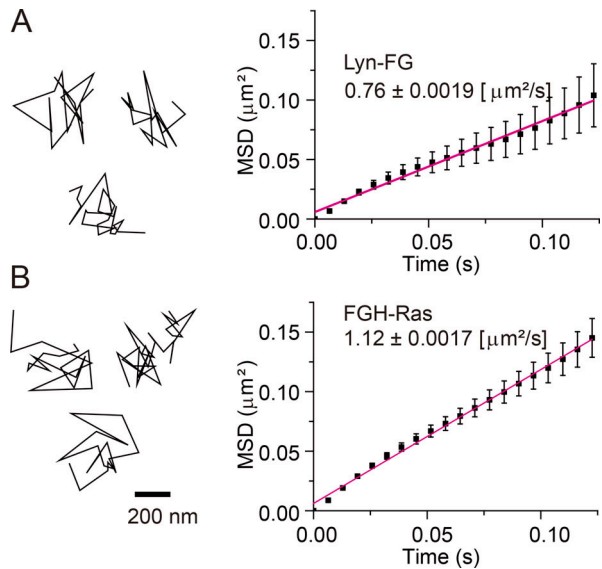

**Figure 4. Lyn-FG and FGH-Ras molecules underwent simple Brownian diffusion in/on the inner PM leaflet as observed at a 6.45-ms resolution, when they were not colocalized with CD59 clusters or Ab-CTXB-GM1 clusters. (A and B)** Representative trajectories of single Lyn-FG (A) and FGH-Ras (B) molecules and the ensemble-averaged MSDs plotted against Δ*t* for Lyn-FG and FGH-Ras. A and B are based on 109 and 456 trajectories, respectively. Their mean diffusion coefficients are shown in the figure. All error bars represent SEM.

clusters. First, we examined the recruitment of myrpal-N20(Lyn)-GFP (Fig. 1 A), which was proposed to be associated with raft domains (Pyenta et al., 2001). The duration histogram for the colocalizations of myrpal-N20(Lyn)-GFP molecules with CD59 clusters clearly exhibited two components (significant difference from *h*(incidental-by-shift), P = 0.025), with a statistically nonsignificant (P < 0.068) 18% reduction in $\tau_2$ compared with the duration histogram for Lyn-FG (Fig. 6 B, a, and Table S1). Second, we found that the TM mutant of Lyn-GFP (TM-Lyn-GFP; Fig. 1 A) did not exhibit any detectable longer-lifetime component in the colocalization duration histogram (Fig. 6 B, b, and Table S1; P = 0.46 against *h*(incidental-by-shift)). These results suggest that (1) the protein moiety of Lyn by itself cannot induce the recruitment; (2) the raft–lipid interaction by itself can induce Lyn recruitment at CD59 clusters; and (3) when both the Lyn protein moiety and raftophilic myristoyl + palmitoyl chains exist, the lifetime at the CD59 cluster raft appears to be prolonged (could be proved in the future when single-molecule imaging is further improved).

To further examine whether the raft–lipid interaction alone can recruit cytoplasmic saturated chain–anchored proteins at CD59 clusters, we examined the recruitment of two more artificial molecules with large deletions in their protein moieties, but with preserved lipid-binding sites: Palpal-N16 growth-associated protein 43 (GAP43)-GFP (raftophilic) and GFP-C5 Rho-gerger (nonraftophilic; Fig. 1 A). Palpal-N16 GAP43-GFP exhibited a clear two-component histogram (significant difference from *h*(incidental-by-shift); P = 0.0023), with a $\tau_2$ (71 ms) quite comparable to the $\tau_2$ values for Lyn-FG and myrpal-N20(Lyn)-GFP with CD59 clusters (Fig. 6 B, c, and Table S1).

Meanwhile, GFP-C5 Rho-gerger did not exhibit any detectable $\tau_2$ component (Fig. 6 B, d, and Table S1; P = 0.97 against *h*(incidental-by-shift)). Taken together, the results obtained with these four designed molecules (Fig. 6 B) suggest that a raft–lipid interaction without a specific protein–protein interaction could induce the recruitment of cytoplasmic proteins with two saturated chains at CD59 clusters. However, if the protein–protein interaction does exist (Lyn-FG; $\tau_2$ = 80 ms), then it could slightly prolong the colocalization lifetime (myrpal-N20(Lyn)-GFP; $\tau_2$ = 66 ms). In short, the outside-in interlayer coupling occurs when stabilized CD59 cluster rafts are induced in the outer leaflet, and the outside-in transbilayer coupling mechanism is predominantly lipid based.

## H-Ras is continually and transiently recruited at CD59 clusters in a manner dependent on raft–lipid interactions

Next, we examined the recruitment of fluorescently labeled H-Ras (FKBP2-GFP-H-Ras [FGH-Ras]), which is anchored to the PM inner leaflet via two saturated (palmitoyl) chains and an unsaturated (farnesyl) chain covalently conjugated to the C-terminal domain of H-Ras (Fig. 1 A). Virtually all of the FGH-Ras molecules underwent thermal diffusion, with a diffusion coefficient (in the time scale of 124 ms) of 1.12 ± 0.0017 μm²/s (Fig. 4 B). The FGH-Ras was functional because it was activated by EGF stimulation (Fig. S2 B).

The histogram of the colocalization durations of FGH-Ras at CD59 clusters exhibited two clear components (Fig. 6 C, a, and Table S1; P = 0.029 against *h*(incidental-by-shift); $\tau_2$ = 91 ms), whereas no significant $\tau_2$ component was detected in the histogram for the colocalizations at nonclustered CD59 (Fig. 6 C, b, and Table S1; P = 0.52 against *h*(incidental-by-shift)). After mildly treating the cells with methyl-β-cyclodextrin (MβCD; 4 mM at 37°C for 30 min), the FGH-Ras colocalization with CD59 clusters was strongly suppressed (Fig. 6 C, c; P = 0.41 against *h*(incidental-by-shift)). The strong effect of partial cholesterol depletion supports the critical importance of raft–lipid interactions for the recruitment of lipid-anchored FGH-Ras at CD59 clusters.

Next, we examined the colocalization of GFP-C10H-Ras-palfar (GFP-tH; Fig. 1 A), which lacks the majority of the H-Ras protein moiety (Prior et al., 2001, 2003), with CD59 clusters. The colocalization duration histogram exhibited two clear components (Fig. 6 C, d, and Table S1; P = 0.027 against *h*(incidental-by-shift); $\tau_2$ = 75 ms versus 91 ms for the full-length FGH-Ras; nonsignificant difference). Taken together, these results suggest that the two palmitoyl chains of H-Ras probably mask the effect of the unsaturated farnesyl chain, and thus FGH-Ras's two palmitoyl chains might work like Lyn-FG's myristoyl + palmitoyl chains. The $\tau_2$ values are summarized in Fig. 6 D.

In the present report, we focused on the recruitment of lipid-anchored cytoplasmic signaling molecules, Lyn-FG and FGH-Ras, at CD59 cluster rafts. Because Lyn-FG and FGH-Ras are continually recruited to CD59 clusters, they are considered to be more concentrated within the nanoscale region (on the order of 10 nm) of the CD59 cluster raft. This will enhance the homo- and heterointeractions of Lyn, H-Ras, and other recruited raftophilic signaling molecules at CD59 cluster rafts. Indeed, we found that

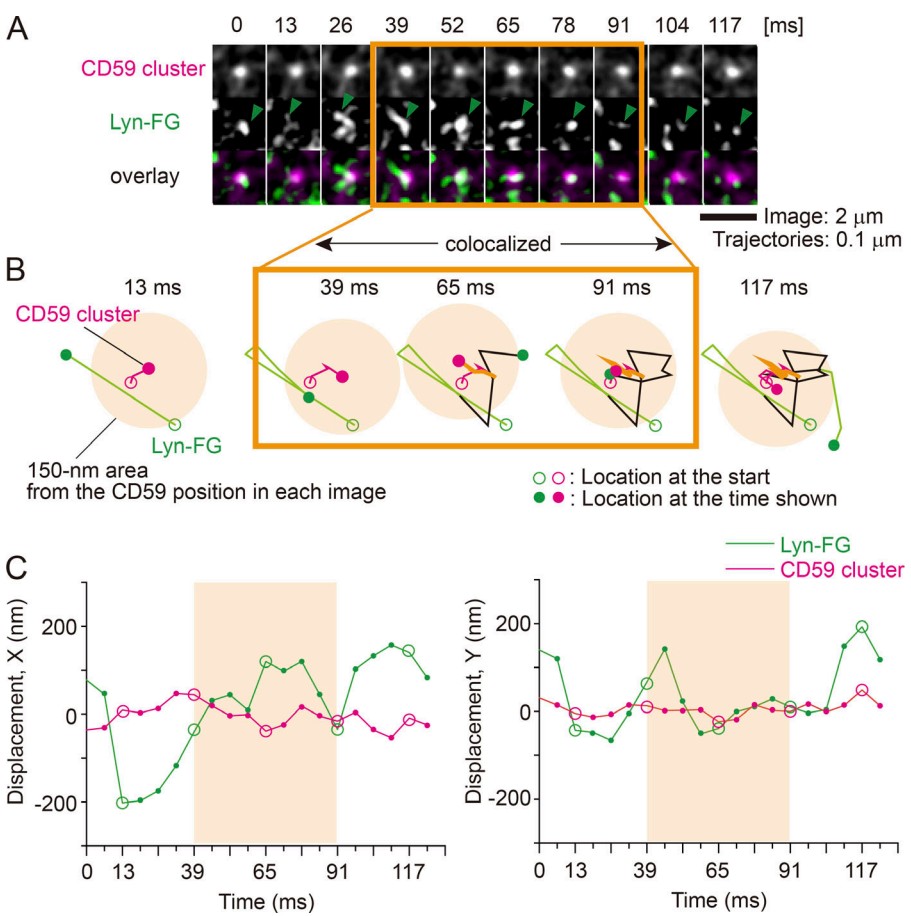

Figure 5. **High-speed, simultaneous, two-color, single-molecule imaging showed transient recruitment of Lyn-FG in/on the inner leaflet at CD59 clusters located in/on the outer leaflet. (A)** Typical single-molecule image sequences (6.45-ms resolution; every other image is shown) showing the colocalization of a CD59 cluster (top row and magenta spots in the bottom row) and a single molecule of Lyn-FG (green arrowheads in the middle row and green spots in the bottom row). Lyn-FG spots appear brighter during colocalization due to slower diffusion. **(B)** The trajectories of the CD59 cluster (magenta) and the Lyn-FG molecule (green) shown in A. These molecules became colocalized (orange circular region with a radius of 150 nm around the CD59 cluster position) between 39 and 91 ms (52 ms; orange box). **(C)** Another display of the colocalization event shown in A and B, showing the displacements of an Lyn-FG molecule and a CD59 cluster along the x and y axes (left and right, respectively) from the average position of the CD59 cluster during the colocalization period, plotted against time. Circles indicate the times employed in B.

the homo-oligomerization of FGH-Ras by cross-linking its FKBP domain by the AP20187 addition could activate FGH-Ras (Fig. S2). This result further suggests that the recruitment of Lyn-FG and FGH-Ras at the small cross-sectional area of the CD59 cluster raft, leading to their higher concentrations at CD59 clusters, would have important signaling consequences.

## GM1 clusters formed by Ab cross-linked cholera toxin B subunit (CTXB) in the PM outer leaflet activate ERK1/2 kinases

To further investigate the raft–lipid interactions across the bilayer, we induced clusters of GM1, a prototypical raft-associated glycosphingolipid (ganglioside), in the PM outer leaflet and examined whether Lyn-FG and FGH-Ras located in/on the inner leaflet could be recruited at GM1 clusters in the outer leaflet. GM1 clusters were induced by applying CTXB conjugated with A633 (dye/protein molar ratio [D/P], 0.8), which could bind five GM1 molecules (CTXB-5-GM1; Merritt et al., 1994), and greater GM1 clusters containing an average of approximately three CTXB and 15 GM1 molecules (virtually 30 saturated acyl chains) were induced by the further addition of a goat polyclonal anti-CTXB Ab IgG (Ab-CTXB-GM1 clusters; Fig. 7 A; see the caption for Fig. 7 B and Materials and methods; the actual variation of the number of CTXB molecules in a greater GM1 cluster could be quite large). We anticipated that all five of the of the GM1 binding sites in CTXB are filled with GM1 because GM1 exists abundantly in the PM outer leaflet of HeLa cells, and the 2D

collision rate is much higher than that in 3D space (Grasberger et al., 1986).

Ab-CTXB-GM1 clusters (30 saturated alkyl chains) diffused with a mean diffusion coefficient of 0.077 µm²/s, 5.1 times slower than non–cross-linked CTXB-5-GM1 (five saturated alkyl chains; 0.39 µm²/s; Fig. 7 D), whereas they diffused 2.6 times slower than CD59 clusters (0.20 µm²/s; 20 saturated alkyl chains; Fig. 2 D). Namely, the average cross-sectional area of the hydrophobic region of the Ab-CTXB-GM1 cluster would be somewhat greater than that of the CD59 cluster.

The GM1 clusters slowly became entrapped in caveolae; ~9% of the fluorescent spots were colocalized with caveolae at 10 min after the addition of the anti-CTXB Abs at 27°C (Fig. S1 B). Therefore, in the present investigation, all of the microscopic observations involving GM1 clusters were made within 10 min after the addition of the Abs, when most of Ab-CTXB-GM1 clusters were located outside caveolae.

CTXB binding to the cell surface did not trigger the ERK signaling cascade, but when Ab-CTXB-GM1 clusters were induced, the ERK signaling cascade was activated (Fig. 3), consistent with the previous observations (Janes et al., 1999; Kiyokawa et al., 2005). The differences found here might be induced by the larger sizes of Ab-CTXB-GM1 clusters than the size of CTXB-5GM1. However, we suspect that this is due not simply to the differences in the sizes of the entire CTXB-5-GM1 and Ab-CTXB-GM1 clusters, but rather to those in the local densities of the

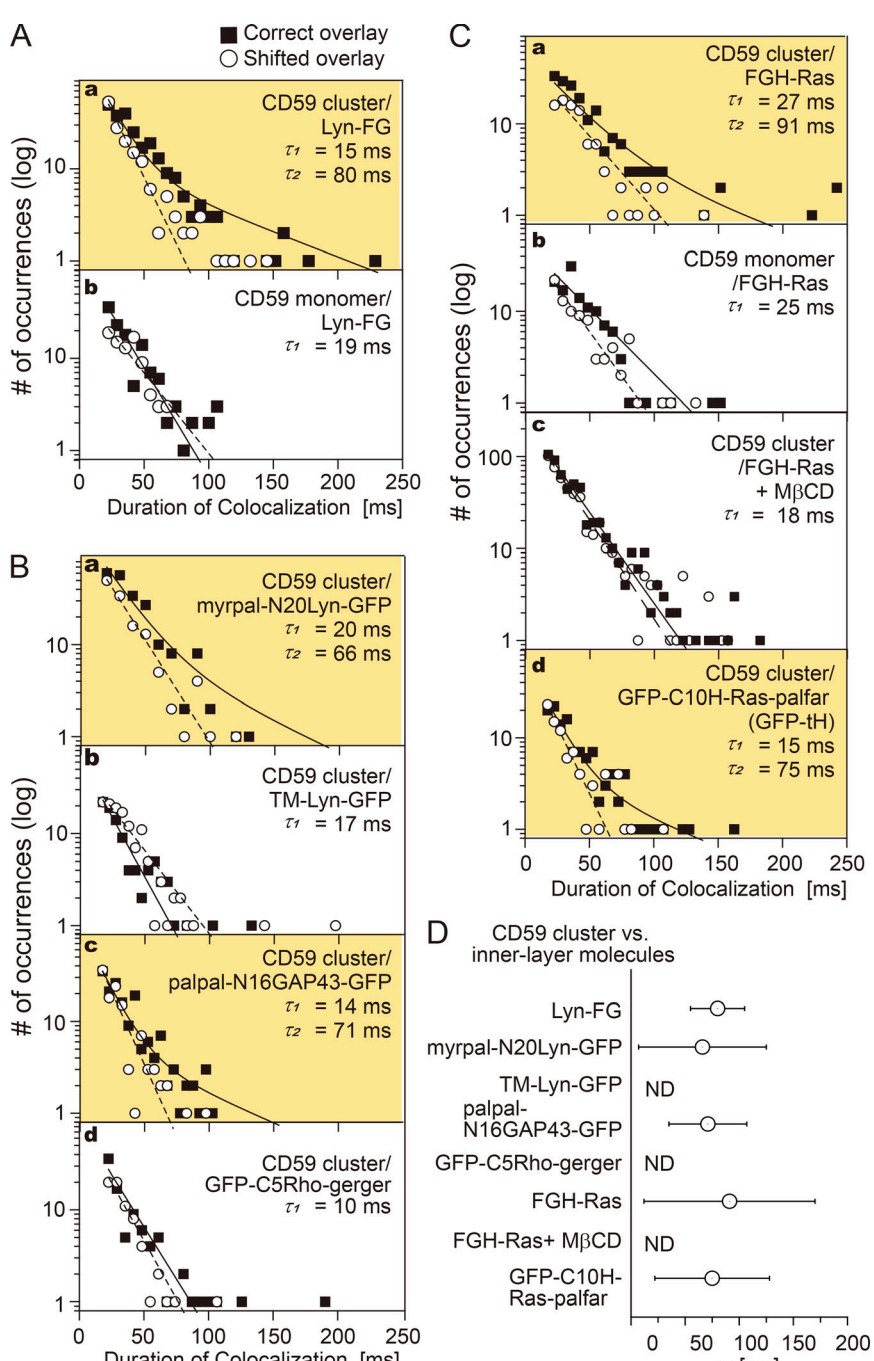

Figure 6. **Lyn-FG, FGH-Ras, and other lipid-anchored raftophilic molecules were recruited at CD59 clusters but not at non–cross-linked CD59.** The distributions (histograms) of the colocalization durations for the "correct" and "shifted" overlays, shown in semilog plots. The histograms for shifted overlays were fitted by a single exponential function (dashed line), and those for the correct overlays were fitted by the sum of two exponential functions (solid line), with the shorter time constant set to $\tau_1$ obtained from the histogram of the shifted overlay. The boxes highlighted in orange contain histograms that could be better fitted with the sum of two exponential decay functions rather than a single exponential function. The values of $\tau_1$ and $\tau_2$ are indicated in each box. See Table S1 for statistical parameters. **(A)** Lyn-FG was recruited at CD59 clusters but not at non–cross-linked CD59 (a, b). **(B)** Recruitment of Lyn-related molecules and other lipid-anchored cytoplasmic model proteins at CD59 clusters: myrpal-N20LynGFP (a) and palpal-N16GAP43-GFP (c) were recruited, but TM-Lyn-GFP (b) and GFP-C5Rhogerger (d) were not. **(C)** FGH-Ras was recruited at CD59 clusters but not at non–cross-linked CD59 (a, b), and FGH-Ras recruitment at CD59 clusters depended on the PM cholesterol (c). Meanwhile, GFP-tH was recruited at CD59 clusters. **(D)** Summary of the bound lifetimes ($\tau_2$) of Lyn-FG, FGH-Ras, and other cytoplasmic lipid-anchored signaling molecules at CD59 clusters. The differences in $\tau_2$ values are nonsignificant. ND, not detected. The MβCD treatments (4 mM at 37°C for 30 min; see part C, c) have been controversial. However, the involvement of raft domains was examined in a variety of methods in the present research, including the use of various lipid-anchoring chains and the TM domain of a prototypical nonraft molecule, LDLR, and a prototypical nonraft phospholipid DOPE. In the past, we employed the MβCD treatments together with other control experiments (using TM artificial mutants of GPI-ARs, saponin treatment, cholesterol repletion after the MβCD treatment) and found that the MβCD treatment with 4 mM MβCD at 37°C for 30 min reproducibly gave the results consistent with the results obtained by using other methods of testing the raft involvement.

saturated chains in the CTXB-5-GM1 and Ab-CTXB-GM1 clusters, based on the following reason.

A crystallographic study showed that the five GM1 binding sites in CTXB are all located ~3.7 nm away from adjacent binding sites (Fig. 1 C; Merritt et al., 1994), and thus the five GM1 molecules (10 saturated chains) are located on a circle with a diameter of ~6.3 nm. Considering the size of the acyl chains (occupying a cross-section of <0.35 nm²; i.e., a diameter of <0.33 nm), two adjacent GM1 molecules bound to CTXB will be located >3 nm away from each other in a space that could accommodate >9 acyl chains and, in the middle of the GM1 binding sites, there would be a circular space with a cross-section of >5 nm in diameter, which could accommodate >25 acyl chains (in the outer leaflet; Fig. 1 C). Namely, the space between two GM1 molecules with saturated acyl chains bound to a CTXB molecule is much larger than the cross-section of a few lipid molecules; that is, CTXB induces only sparse GM1 clusters. The observation that CTXB molecules simply bound to the PM cannot trigger the downstream signals is consistent with this consideration: the five GM1 molecules bound to a single CTXB molecule would not provide the threshold densities of saturated lipids necessary to create stable rafts by assembling and keeping cholesterol and saturated chains in CTXB-5-GM1 and excluding unsaturated chains. The five GM1 molecules bound to a single CTXB molecule would not serve as a nucleus to induce raft domains beneath the CTXB molecule (in the outer leaflet of the PM) and hence would

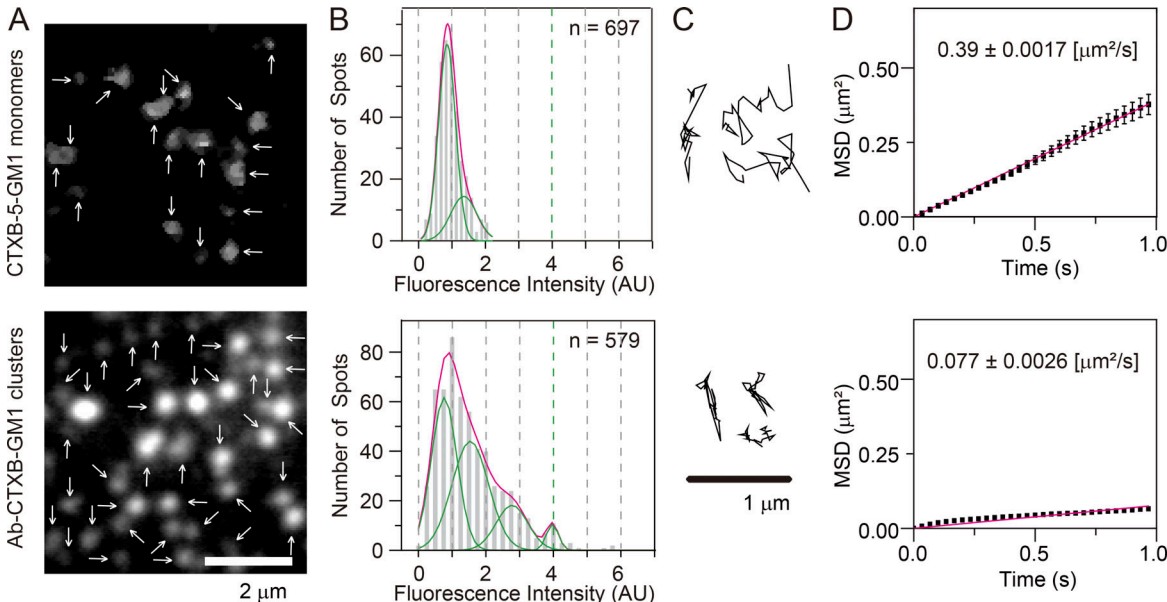

Figure 7.  **Ab-CTXB-GM1 clusters generated in the PM outer leaflet contained an average of ~15 GM1 molecules and diffused 2.6 times slower than CD59 clusters. (A)** Fluorescence images of non–cross-linked fluorescently labeled CTXB (which could bind up to five GM1 molecules; A633 conjugated with a D/P of 0.80; called CTXB-5-GM1; top) and CTXB clusters induced by the further addition of anti-CTXB Abs (Ab-CTXG-GM1 cluster; bottom), obtained at single-molecule sensitivities. Arrows indicate all of the detected fluorescence spots in each image. **(B)** Histograms showing the distributions of the signal intensities of individual fluorescent spots of A633-labeled CTXB-5-GM1 (top) and Ab-CTXB-GM1 clusters (bottom). On the basis of these histograms, we concluded that each Ab-CTXB-GM1 cluster contained an average of ~15 GM1 molecules (see Materials and methods), although the distribution would be quite broad. **(C)** Typical trajectories of CTXB-5-GM1 (top) and Ab-CTXB-GM1 clusters (bottom) for 0.2 s, obtained at a time resolution of 6.45 ms. **(D)** Ensemble-averaged MSDs plotted against time, suggesting that in the time scale of 1 s, both CTXB-5-GM1 (top; 154 trajectories) and Ab-CTXB-GM1 clusters (bottom; 91 trajectories) undergo effective simple Brownian diffusion, and the diffusion is slowed by a factor of about 5 after Ab-induced clustering. All error bars represent SEM.

fail to recruit signaling molecules that trigger the ERK signaling cascade. This possibility was directly examined in the present study (see the next section; in the case of CD59 clusters, we suspect that due to the long flexible glycochain of GPI, CD59 has reorientation freedom, and thus the saturated chains of CD59 in the cluster and the cholesterol, sphingomyelin, and gangliosides recruited from the bulk PM can form a tighter complex beneath the cluster of CD59 protein moieties).

Meanwhile, when the CTXB molecules were cross-linked by anti-CTXB Abs, because the GM1 molecules are bound near the outer edges of CTXB (Merritt et al., 1994), they would be located very close to the GM1 molecules bound to other CTXB molecules in the Ab-CTXB-GM1 cluster (Fig. 1 C). These closely associated GM1 molecules could form the stable raft nucleus for recruiting cholesterol and lipids with saturated alkyl chains, recruiting raftophilic signaling molecules and thus triggering the ERK signaling pathways.

The stabilization and enlargement of raft domains induced by CTXB and its Abs as well as signaling by the enhanced raft domains have been established quite well in the literature, although the data have been quite qualitative (reviewed by Kusumi et al., 2020). For example, using the T cell line E6.1 Jurkat, Janes et al. (1999) reported that the addition of CTXB and its Ab-induced membrane patches contained lymphocyte-specific protein tyrosine kinase (Lck), linker for activation of T cells (LAT), and the T cell receptor, but excluded CD45. These patches were considered to be enhanced raft domains because they were colocalized by CD59, used as a prototypical raft marker. Therefore, we next investigated whether

CTXB-5-GM1 or Ab-CTXB-GM1 clusters could recruit Lyn-FG and FGH-Ras.

**Lyn and H-Ras are continually and transiently recruited at Ab-CTXB-GM1 clusters in a manner dependent on raft–lipid interactions, but not at CTXB-5-GM1**

We directly examined whether single molecules of Lyn-FG and FGH-Ras were recruited at CTXB-5-GM1 and Ab-CTXB-GM1 clusters located in/on the PM outer leaflet. As described in the previous section, CTXB-5-GM1 failed to trigger ERK activation, in contrast to Ab-CTXB-GM1 clusters.

The histogram of the colocalization durations of FGH-Ras at Ab-CTXB-GM1 exhibited two clear components, indicating that Lyn-FG was recruited at Ab-CTXB-GM1 (Fig. 8 A, a, and Table S2; P = 0.013 against $h$(incidental-by-shift); $\tau_2$ = 110 ms). Meanwhile, no significant $\tau_2$ component was detectable for the colocalizations at CTXB-5-GM1 (Fig. 8 A, b, and Table S2; P = 0.24 against $h$(incidental-by-shift)). Similarly, FGH-Ras was recruited at Ab-CTXB-GM1 (Fig. 8 B, a, Table S2; P = 0.025 against $h$(incidental-by-shift); $\tau_2$ = 97 ms), but not at CTXB-5-GM1 (Fig. 8 B, b, and Table S2; P = 0.52 against $h$(incidental-by-shift)).

Partial cholesterol depletion eliminated the $\tau_2$ component for the FGH-Ras colocalization with Ab-CTXB-GM1 (Fig. 8 B, c, and Table S2; P = 0.96 against $h$(incidental-by-shift)). Furthermore, when DNP-DOPE, a nonraft reference unsaturated phospholipid, was clustered in the outer leaflet by the addition of anti-DNP Abs and secondary Abs (the Ab concentrations were adjusted so that >90% of DNP-DOPE clusters became immobile; i.e., the cross-section

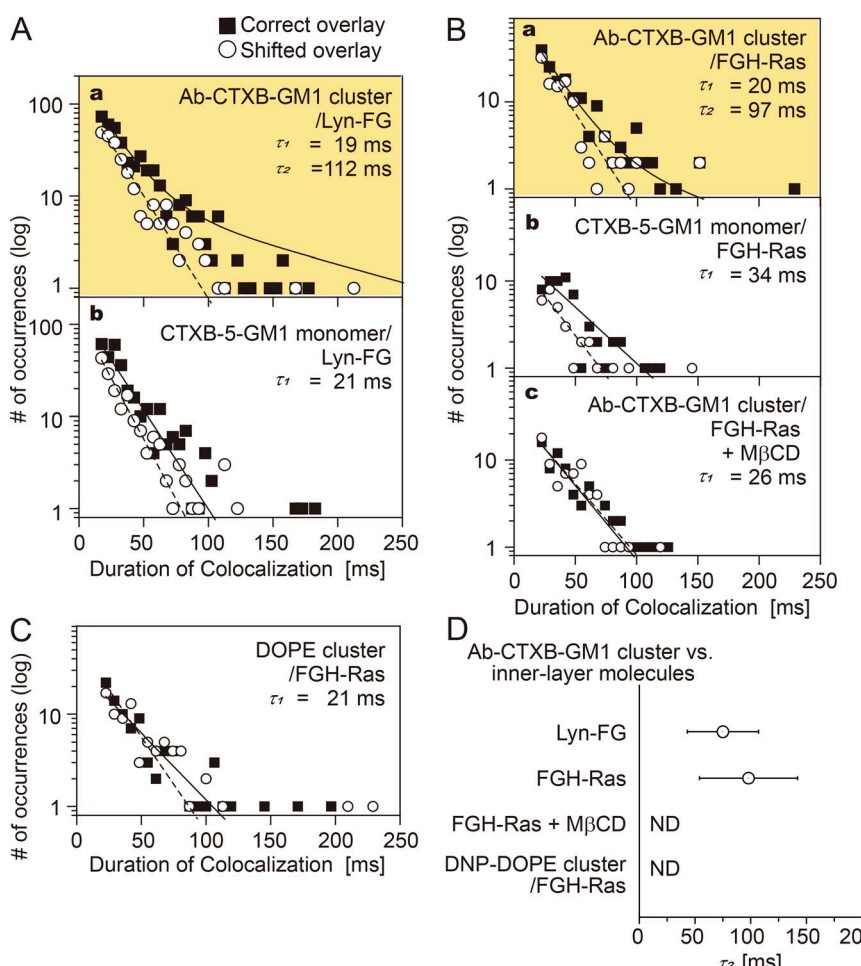

Figure 8. **Lyn-FG and FGH-Ras were recruited at Ab-CTXB-GM1 clusters but not at CTXB-5-GM1.** The distributions (histograms) for the colocalization durations are shown. See the Fig. 6 legend for details and keys. See Table S2 for statistical parameters. **(A)** Lyn-FG was recruited at Ab-CTXB-GM1 clusters but not at CTXB-5-GM1 (a, b). **(B)** FGH-Ras was recruited at Ab-CTXB-GM1 clusters but not at CTXB-5-GM1 (a, b), and its recruitment at Ab-CTXB-GM1 clusters depended on the PM cholesterol (c). **(C)** FGH-Ras was not recruited to DNP-DOPE clusters. **(D)** Summary of the bound lifetimes ($\tau_2$) of Lyn-FG and FGH-Ras at Ab-CTXB-GM1 clusters. ND, $\tau_2$ component not detected.

of the DNP-DOPE cluster would be substantially greater than that of Ab-CTXB-GM1), no significant $\tau_2$ component was detected for the FGH-Ras colocalization with DNP-DOPE clusters (Fig. 8 C and Table S2; P = 0.39 against $h$(incidental-by-shift)). These results further support the proposal that raft–lipid interactions are essential for the recruitment of cytoplasmic lipid-anchored signaling molecules at Ab-CTXB-GM1 and therefore that the GM1 molecules closely apposed to each other inside the Ab-CTXB-GM1 cluster induce stable raft nuclei by recruiting cholesterol and other raftophilic molecules. The results for $\tau_2$ are summarized in Fig. 8 D.

### Small clusters of inner-leaflet signaling molecules did not recruit CD59 or GM1 in the outer leaflet

The homo-oligomerization of Lyn-FG and FGH-Ras in the cytoplasm was induced by the addition of AP20187 (dimerizer system developed by Schreiber and then ARIAD Pharmaceuticals; Schreiber, 1991; Clackson et al., 1998). The presence of a single FKBP molecule in a protein could only create dimers but not oligomers greater than dimers upon AP20187 addition, but the presence of two FKBP molecules in a single protein could induce oligomers (Fig. 1 B, bottom). The average number of Lyn-FG or FGH-Ras molecules in a single cluster was estimated to be approximately three (Fig. S2, C and D, and Materials and methods).

The oligomerized FGH-Ras triggered the downstream signaling, as shown by the pull-down assay using the Ras-binding domain of the downstream kinase Raf-1 (Fig. S2 B), consistent with previous observations (Inouye et al., 2000; Nan et al., 2015). Meanwhile, the oligomerization-induced self-phosphorylation of Lyn-FG was not detected (Fig. S2 A).

We examined whether CD59 and CTXB-5GM1 located in the PM outer leaflet could be recruited at FGH-Ras or Lyn-FG oligomers induced in the PM inner leaflet by the addition of AP20187 (Fig. 9 and Table S3). No significant recruitment was detectable, indicating that the oligomers of the inner-leaflet lipid-anchored signaling molecules cannot recruit the outer-leaflet raft-associated molecules. This result suggests that although FGH-Ras and Lyn-FG could be transiently recruited to stabilized raft domains, they would only be passengers and not the main molecules for inducing raft domains, probably due to their shorter saturated chains (palmitoyl) and the presence of unsaturated chains. Furthermore, FGH-Ras and Lyn-FG could only be recruited to the outer edges of the raft domains or perhaps the interfaces of the raft and bulk domains. Meanwhile, the lack of CD59 and GM1 recruitment might be due to the smaller sizes (an average of approximately three molecules) of the FGH-Ras and Lyn-FG oligomers.

## Discussion
The recruitment of cytoplasmic signaling molecules to small regions in the PM after stimulation is considered to be important

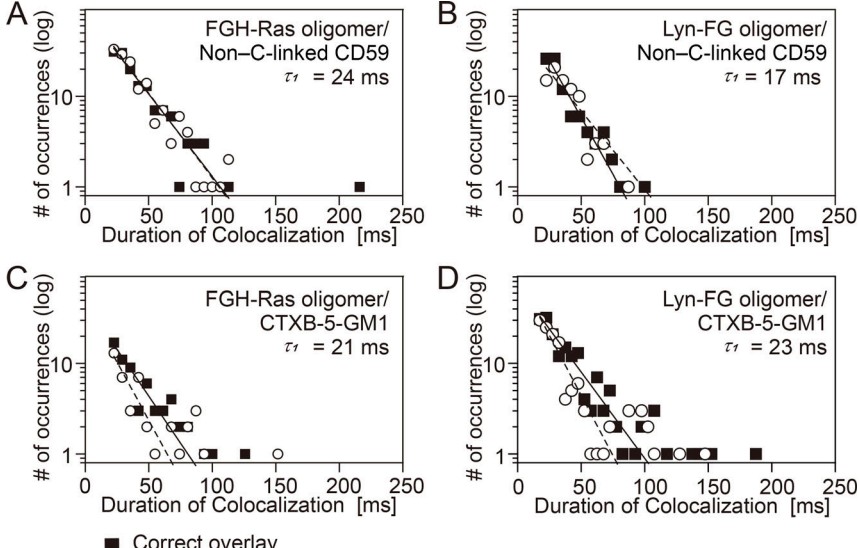

Figure 9. **FGH-Ras oligomers and Lyn-FG oligomers induced by AP20187 addition failed to recruit non–cross-linked CD59 and CTXB-5-GM1.** Shown here are the histograms for the durations in which non–cross-linked CD59 and CTXB-5-GM1 located in/ on the PM outer leaflet are colocalized with FGH-Ras oligomers and Lyn-FG oligomers artificially induced in the PM inner leaflet by the addition of AP20187. See the Fig. 6 legend for details and keys. See Table S3 for statistical parameters. **(A)** Recruitment of non–cross-linked CD59 located in the outer leaflet at the induced FGH-Ras oligomers located in the inner leaflet. **(B)** Recruitment of non–cross-linked CD59 located in the outer leaflet at the induced Lyn-FG oligomers located in the inner leaflet. **(C)** Recruitment of CTXB-5-GM1 located in the outer leaflet at the induced FGH-Ras oligomers located in the inner leaflet. **(D)** Recruitment of CTXB-5-GM1 located in the outer leaflet at the induced Lyn-FG oligomers located in the inner leaflet.

for inducing the downstream signaling, because higher concentrations of signaling molecules in small regions would enhance homo- and heterointeractions and possibly the formation of transient dimers and oligomers. We indeed found that the homo-oligomerization of FGH-Ras induced by AP20187 activated the downstream signaling of FGH-Ras and H-Ras (Fig. S2 B). Therefore, in the present research, we extensively studied the recruitment of Lyn, H-Ras, and other lipid-anchored cytoplasmic molecules at CD59 cluster rafts and Ab-CTXB-GM1 clusters.

Our results clearly showed that Ab-CTXB-GM1 clusters of a raftophilic lipid (GM1) formed in the PM outer leaflet can recruit the cytoplasmic lipid-anchored signaling molecules Lyn and H-Ras to the inner-leaflet region apposed to the outer-leaflet Ab-CTXB-GM1 clusters. This recruitment was not induced after the PM cholesterol was mildly depleted or when the unsaturated lipid (DNP)-DOPE was clustered in the outer leaflet. These results unequivocally demonstrate that the cytoplasmic lipid-anchored signaling molecules Lyn and H-Ras can be assembled at the stabilized raft–lipid clusters formed in the outer leaflet by raft–lipid interactions. The involvement of TM proteins in the recruitment process would be quite limited, because (1) even cytoplasmic lipid-anchored molecules after the deletions of the majorities of their protein moieties were recruited at Ab-CTXB-GM1 clusters, (2) their dwell lifetimes at CD59 clusters and Ab-CTXB-GM1 clusters were very similar to those of Lyn-FG and FGH-Ras, and (3) the recruitment of FGH-Ras depended on the PM cholesterol level. Of course, this does not rule out the specific interactions of GPI-ARs with TM proteins as coreceptors (Klein et al., 1997; Wang et al., 2002; Zhou, 2019). The results showing that Ab-CTXB-GM1 clusters, but not CTXB-5GM1, can recruit Lyn-FG and FGH-Ras (Fig. 8) would suggest that critical concentrations (number densities) of saturated chains would probably exist for generating the outer-leaflet raft domains that can recruit raftophilic molecules in the inner leaflet. However, the concentration effect might further be compounded by the larger sizes of the Ab-CTXB-GM1 cluster-induced raft domains compared with the CTXB-5GM1–induced raft domains.

To summarize the sequence of events in CD59 signaling (Fig. 10), first, the stable CD59 cluster rafts in the outer leaflet are induced by the clustering of raftophilic CD59 molecules by extracellular stimulation, such as MACC binding. The stabilized raft domains tend to last for durations on the order of tens of minutes (Suzuki et al., 2007b, 2012), whereas their constituent molecules, such as the gangliosides, tend to stay there only for 50 ms and turn over quickly, continually exchanging with those located in the bulk PM region (Komura et al., 2016). Second, at the signal-induced stabilized CD59 cluster raft domains, the raftophilic cytoplasmic signaling molecules, Lyn and H-Ras, are recruited by raft–lipid interactions with lifetimes on the order of 0.1 s (Figs. 6 and 8); that is, each molecule stays at the CD59 cluster raft quite transiently. However, because many molecules would continually arrive one after another, and because each raft domain can accommodate several hundred lipid molecules (when the raft radius is 10 nm, each leaflet within the raft can accommodate ~500 phospholipids), many cytoplasmic raftophilic signaling molecules could be dynamically concentrated in the small cross-sectional area beneath the CD59 cluster raft, leading to locally enhanced molecular interactions.

Let us assume that the sizes of the stabilized CD59 cluster rafts and Ab-CTXB-GM1 cluster rafts are in the range of 20–100 nm in diameter (Figs. 1, B and C; 2; and 7) and the diffusion coefficient of the lipid-anchored signaling molecules is ~1 μm²/s (Fig. 4). Then, these signaling molecules would stay in the 20–100-nm region in the bulk PM for only 0.03–0.63 ms. However, they remained in the stabilized raft domains for 80–110 ms (Figs. 6 and 8); that is, the dwell lifetimes were prolonged by a factor of 200–2,000, which is a large factor. Namely, the dwell lifetimes in the range of 80–110 ms might appear to be short, but in fact, Lyn-FG, FGH-Ras, and other raftophilic lipid-anchored molecules exhibited extremely prolonged dwell lifetimes beneath the stabilized raft domains in the outer leaflet. Such extreme prolongation would not be possible by simple interactions of the lipids in the inner leaflet with the lipids in stabilized raft domains in the outer leaflet.

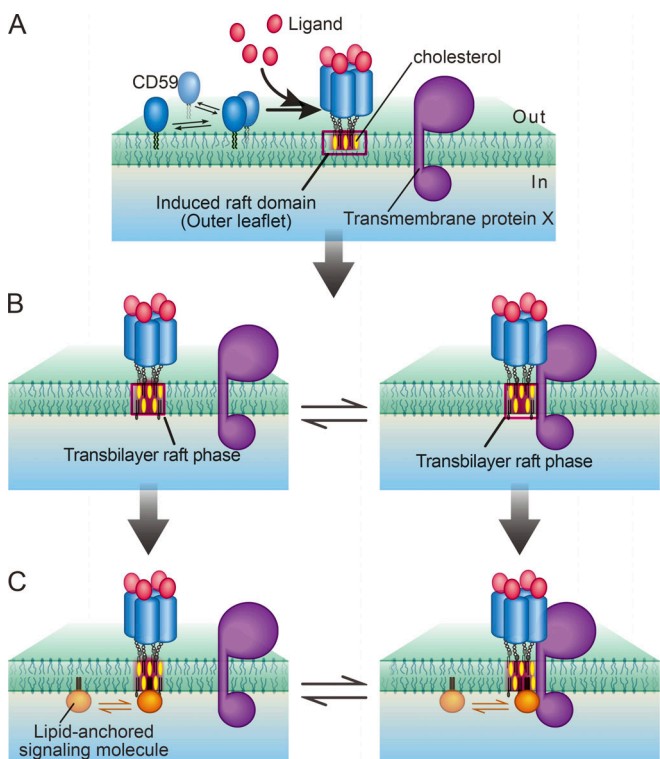

Figure 10. **Schematic model showing the CD59 signal transduction mediated by the transbilayer raft phase, which recruits lipid-anchored signaling molecules at the ligated, stabilized CD59 cluster domains in the PM outer leaflet, inducing enhanced interactions of recruited molecules. (A)** First, the ligand binding triggers the conformational changes of CD59, which in turn induce CD59 clustering, creating stable CD59 cluster signaling rafts. If GM1 is clustered closely, then stable GM1 cluster rafts will be produced. **(B)** Then, the transbilayer raft phase is induced by the CD59 cluster raft by involving molecules in the inner leaflet, recruiting cholesterol and molecules with saturated alkyl chains (left) and also excluding molecules with unsaturated alkyl chains. An as yet unknown TM protein(s) X, which has affinities to raft domains, might also be recruited to the transbilayer raft phase (right; recruitment of X could be enhanced by specific protein–protein interactions with the ligated CD59 exoplasmic protein domain). **(C)** Finally, cytoplasmic lipid-anchored signaling molecules, such as H-Ras and Lyn, are recruited to the transbilayer raft phase in the inner leaflet by the raft–lipid interaction (left). This could be enhanced by the protein–protein interaction with the TM protein X (right). Although the residency times of the inner-leaflet signaling molecules beneath the CD59 cluster may be limited, because many molecules will be recruited there one molecule after another, interactions of two or more species of cytoplasmic signaling molecules will occur efficiently beneath the CD59 cluster raft. This way, the transbilayer raft phase induced by the stabilized CD59 cluster raft would function as an important signaling platform.

Because the micrometer-scale transbilayer raft phases have been detected in artificial bilayer membranes (Collins and Keller, 2008; Blosser et al., 2015), we propose that nanoscale transbilayer raft phases are induced in both leaflets by the stabilized raft domains initially formed in the outer leaflet and that when cytoplasmic raftophilic lipid-anchored signaling molecules arrive at the transbilayer raft phases, they tend to be trapped in the inner-leaflet part of the transbilayer raft phase, exhibiting dwell lifetimes on the 0.1-s order. Namely, transbilayer raft–lipid interactions would only make sense when

cooperative lipid interactions occur due to the formation of the transbilayer raft phase. Indeed, raft domains are generally considered to form by the cooperative interactions of saturated alkyl chains and cholesterol, as well as by their cooperative exclusions from the bulk PM enriched in unsaturated alkyl chains. Therefore, we propose that the nanoscale transbilayer raft phases would be induced at stabilized rafts initially formed in the outer leaflet. We further propose that the transbilayer raft phases induced by the stimulation-triggered GPI-AR cluster rafts would act as a key signaling platform for the engaged GPI-AR clusters, recruiting inner-leaflet raftophilic signaling molecules, and that the formation of the transbilayer raft phase would be a general mechanism for GPI-AR signal transduction (however, this does not rule out the possibility that some TM proteins with raft affinities or those that can be concentrated at the interface between the raft and bulk domains are involved in the recruitment of Lyn and H-Ras, as depicted in Fig. 10; compare the result shown in Fig. 6 A, a, with that shown in Fig. 6 B, a).

This recruitment mechanism based on the transbilayer raft phase appears to suggest the lack of specificity in the cytoplasmic signaling without any dependence on the GPI-AR species. However, because different GPI-ARs would form signaling cluster rafts with a variety of sizes, because of closeness of the saturated acyl chains within the cluster (as found here for GM1 clusters induced by CTXB), and because the TM protein species with which different GPI-ARs interact would vary (Suzuki et al., 2012; Zhou, 2019), the GPI-AR cluster rafts formed in the outer leaflet could induce transbilayer raft phases with distinct properties. These transbilayer raft phases could recruit a variety of lipid-anchored signaling molecules with differing efficiencies, thus triggering various downstream signaling cascades with different strengths; that is, the relative activation levels among the many intracellular signaling cascades triggered by GPI-ARs would vary depending on the GPI-AR species.

## Materials and methods

### Improved camera systems for simultaneous, dual-color, single-molecule imaging in living cells at enhanced time resolutions of 5 and 6.45 ms

The major improvement of our single-molecule imaging station from the previously published version (Koyama-Honda et al., 2005; Komura et al., 2016; Kinoshita et al., 2017) was the employment of two camera systems that allow higher frame rates. With an increase in the frame rate of the camera system, we employed lasers with higher outputs (see the next paragraph). The two camera systems both employed two-stage microchannel plate intensifiers (C8600-03; Hamamatsu). In one camera system, the image intensifier was lens coupled to an electron multiplying charge-coupled device camera (Cascade 650; Photometrics), which was operated at 155 Hz (6.45 ms/frame), with a frame size of 653 × 75 pixels (38.9 × 4.46 μm² for a total of 240× magnification). In the other camera system, the image intensifier was fiber coupled, with a 1.6:1 tapering, to a charge-coupled device camera (XR/MEGA-10ZR; Stanford Photonics) cooled to –20°C and operated at 200 Hz (5 ms/frame), with a frame size of 640 × 160 pixels (27.1 × 6.75 μm² for a total of 240× magnification).

Right before microscopic observations of the cells, the culture medium was replaced by HBSS buffered with 2 mM Pipes at pH 7.4 (P-HBSS), and the bottom PMs of the cells growing on glass-bottom dishes were observed by a homebuilt objective lens–type total internal reflection fluorescence microscope constructed on an inverted microscope (IX-70; Olympus) with a 60× objective lens (NA, 1.4) with two detection arms for simultaneous two-color single-molecule imaging, as described previously (Koyama-Honda et al., 2005). The temperature of the sample and the microscope was maintained at 27 ± 1°C. The cells were illuminated simultaneously by a 488-nm laser (for GFP, Sapphire 488-20; Coherent) and a 594-nm laser (for A633, 05-LYR-173; Melles Griot/IDEX Health & Science). Fluorescence signals from GFP and A633 were split into the two detection arms by using a dichroic mirror at 600 nm (600DCXR; Chroma) and further isolated by interference filters (HQ535/70 for GFP and HQ655/100 for A633; Chroma). The fluorescence image in each arm was projected onto the photocathode of the image intensifier in the camera system described above (the same cameras were employed for the two channels). MetaMorph software (Molecular Devices) was used for image acquisition and preprocessing, and the obtained images were further processed using ImageJ software.

### Determining the positions of fluorescence spots of single molecules and molecular clusters in the image

The positions (x and y coordinates) of individual fluorescence spots were determined by using an in-house computer program (Koyama-Honda et al., 2005; Hiramoto-Yamaki et al., 2014; Fujiwara et al., 2016), based on a spatial cross-correlation matrix (Gelles et al., 1988). For each frame, the entire image was correlated with a symmetric 2D Gaussian point spread function with an SD of 150 nm (kernel). The resulting 2D cross-correlation function for each molecule and each molecular cluster was thresholded, and their positions were determined as the center of mass of the thresholded correlation intensity.

### Colocalization detection and evaluation of colocalization lifetimes

For the colocalization analysis, GFP trajectories longer than 19 frames and A633 trajectories longer than 29 frames were used. The colocalization of an A633 spot with a GFP spot was defined as the event in which the two fluorescence spots, representing A633 and GFP molecules, became localized within 150 nm of each other. This is a distance at which an exactly colocalized molecule is detected as colocalized at probabilities >90%, using the Cascade 650 camera operated at 155 Hz, and higher probability was achieved using the XR/MEGA-10ZR camera operated at 200 Hz (Koyama-Honda et al., 2005).

A colocalization distance of 150 nm is much greater than the molecular scale, and therefore, in addition to colocalization due to specific molecular binding, events in which molecules incidentally encounter each other within a distance of 150 nm, termed "incidental colocalizations," can occur. However, as described in the Results section, nonassociated molecules may track together by chance over a short distance, but the probability of moving together for multiple frames is small, and

therefore longer colocalizations imply the binding of two molecules.

In the analysis of colocalization durations, those as short as one or two frames were neglected to avoid higher-frequency noise. Likewise, if two colocalization events are separated by a gap of one or two frames, then they are linked and counted as a single longer colocalization event. To obtain the histogram of incidental colocalization durations, the image obtained in the longer-wavelength channel (A633) was shifted toward the right by 20 pixels (1.0 and 1.19 μm, depending on the camera) and then overlaid on the image obtained in the GFP channel ("shifted overlay"). The histogram of the incidental colocalization durations was called $h$(incidental-by-shift). We found $h$(incidental-by-shift) could effectively be fitted by a single exponential decay function, using nonlinear least-squares fitting by the Levenberg-Marquardt algorithm provided in OriginPro software, and the decay time constant was called the "incidental colocalization lifetime," $\tau_1$ (Figs. 6 and 8).

Meanwhile, the distribution of the colocalization durations for correctly overlaid A633 and GFP images ("correct overlay") was obtained, and we found that some of the histograms (such as that for Lyn-FG versus CD59 clusters) could be fitted with the sum of two exponential functions with a decay time constant $\tau_1'$ and the other, longer time constant $\tau_2$ (Figs. 6 and 8). The $\tau_1'$ component was considered to represent the duration of incidental colocalization, and thus $\tau_1' = \tau_1$. Therefore, in the following discussion, we describe $\tau_1'$ simply as $\tau_1$.

The $\tau_2$ component of the histogram was considered to describe the colocalization durations, including the durations of true molecular interactions ($\tau_B$). Here, we propose that the binding duration $\tau_B$ can be approximated by $\tau_2$, which can be directly determined from the histogram, based on the following argument. As described previously (Kasai et al., 2018), in the simplest and probably most primary case in which the binding occurs only once during a single colocalization event, the duration $\tau_2$ would be the sum of (1) the duration between the incidental encounter and actual molecular binding, (2) the duration of molecular binding ($\tau_B$), and (3) the duration between the dissociation of two molecules and separation by >150 nm. Therefore, the mathematical function to describe the histogram for the colocalization durations including the molecular binding would be $\exp(-t/\tau_B)$ convoluted with the histogram $h$(incidental-by-shift), which is proportional to $\exp(-t/\tau_1; t = $ time) at the present experimental accuracies (see, e.g., Sungkaworn et al., 2017; Figs. 6 and 8). Here, we are assuming simple zero-order kinetics for the release of lipid-anchored cytoplasmic molecules from the CD59 cluster rafts (and thus the binding duration distribution is proportional to $\exp(-t/\tau_B)$). The result of the convolution of an exponential function with another exponential function is well known, and the convoluted function is the sum of these two exponential functions ($\exp(-t/\tau_1)$ and $\exp(-t/\tau_B)$). Therefore, the entire histogram is the sum of the histogram for simple close encounters, $h$(incidental-by-shift), which has the form of $\exp(t/\tau_1)$, and the histogram for the colocalization events that include molecular interactions and is expressed by the sum of $\exp(-t/\tau_1)$ and $\exp(-t/\tau_B)$. Meanwhile, as described, some of the experimentally obtained histograms (such as that

for Lyn-FG versus CD59 clusters) could be fitted with the sum of two exponential functions with the decay time constant $\tau_1$ and the other, longer time constant $\tau_2$ (Figs. 6 and 8). Therefore, we find $\tau_B = \tau_2$. Namely, the longer time constant $\tau_2$ obtained from the fitting represents the binding duration (Figs. 6 D and 8 D).

For the actual two-component fitting for the histograms of the correctly overlaid images, the exponential lifetime for the faster decay function was fixed at the $\tau_1$ value determined from the histogram of the shifted overlay $h$(incidental-by-shift), and then the fitting with the sum of two exponential functions was performed. For some intracellular signaling molecules, the second component was undetectable, indicating that the colocalization did not take place. Throughout this report, the Brunner-Munzel test was used for the statistical analysis, and its result and the mean, SEM, the number of conducted experiments, and all other statistical parameters are summarized in Table S1, Table S2, and Table S3.

However, due to the problem of the signal-to-noise ratios, the actual estimation of $\tau_2$ involved quite large errors. Accordingly, in the present study, we paid more attention to whether the duration histogram could be represented by a single exponential decay function or the sum of two exponential decay functions.

Note the following. When two molecules become colocalized within the 150-nm radius area, in general, the actual binding can occur multiple times before they become separated farther than 150 nm, prolonging the colocalized durations. The Brownian simulation and theory predict that, even in these general cases, the distribution of the colocalized durations could be described by the sum of two exponential functions (Redner, 2001), and in the case in which the time resolution is not sufficient, their decay time constants will be given by $\tau_1$ and $\tau_2$ employed here (i.e., the observed $\tau_2$ component is dominated by the duration of one-time binding, as we assumed). Therefore, by assuming that the incidental colocalization lifetimes could be approximated by a single exponential function, $\exp(-t/\tau_1)$, the final functional form (the addition of two exponential functions) should be able to describe the experimental histograms quite well.

### Plasmid generation
The cDNA encoding two tandem FKBPs (FKBP2) was obtained from the pC4-Fv1E vector (ARGENT Regulated Homodimerization Kit; ARIAD Pharmaceuticals) and subcloned into the pTRE2hyg vector (including a tetracycline [Tet]-responsive element promoter; Takara Bio) with the cDNA encoding GFP-H-Ras (a kind gift from A. Yoshimura, Keio University School of Medicine, Tokyo, Japan; Murakoshi et al., 2004) to produce FGH-Ras. The cDNA encoding Lyn was obtained from RBL-2H3 cells and subcloned into the pTRE2hyg vector with the cDNA encoding FKBP2 and EGFP (derived from pEGFP-N2; Clontech/Takara Bio) to produce Lyn-FKBP2-GFP (Lyn-FG). The cDNA encoding EGFP was subcloned with the signal sequence 5′-GGG TGCCTTGTCTTGTGA-3′ for the geranylgeranyl modification (CAAX) into the pTRE2hyg vector to produce GFP-C5 Rho-gerger. The cDNAs encoding myrpal-N20Lyn-GFP, Palpal-N16GAP43-GFP, and GFP-tH were constructed as described previously (Pyenta et al., 2001; Zacharias et al., 2002; Prior et al., 2003). The cDNA encoding TM-Lyn-GFP was generated by

linking the cDNA sequence for the signal peptide derived from the LDLR to the T7-tag sequence, the TM domain of the LDLR sequence, the cDNA encoding Lyn with a deletion of the N-terminal six aa (myrpal modification site), then to the GFP sequence, and subcloning the produced cDNA sequence into the pTRE2hyg vector. The cavelin-1–GFP vector and GST–Rho-binding domain (GST-RBD) vector were generous gifts from T. Fujimoto (Nagoya University School of Medicine, Nagoya, Japan; Kogo and Fujimoto, 2000) and A. Yoshimura (Murakoshi et al., 2004), respectively.

### Cell culture, transfection, and expression of chimeric molecules
HeLa Tet-Off cells and Tet-On cells (Clontech/Takara Bio) were maintained in MEM (Life Technologies) supplemented with 10% FBS (MilliporeSigma) and transfected with each plasmid using Lipofectamine Plus (Life Technologies). HeLa Tet-Off cells stably expressing FGH-Ras and HeLa Tet-On cells stably expressing Lyn-FG, myrpal-N20Lyn-GFP, TM-Lyn-GFP, Palpal-N16GAP43-GFP, GFP-C5 Rho-gerger, and GFP-tH were selected in medium containing 0.2 mg/ml hygromycin, and positive clones were captured with micropipettes. The vector encoding cavelin-1–GFP was transfected using Lipofectamine Plus, and the protein was transiently expressed in HeLa Tet-On cells. Before single-molecule observations, HeLa cells were replated on 12-mm-diameter glass-bottom culture dishes (Iwaki) and cultured for 2–3 d. The medium for the FGH-Ras–expressing HeLa Tet-Off cells contained 2 µg/ml doxycycline (Dox; ICN Biomedicals) to reduce the expression of recombinant molecules to levels suitable for single-molecule observations. The medium for Tet-On cells expressing GFP fusion proteins did not contain Dox, because, even without Dox-induced expression, the expression levels were sufficiently high for single-molecule observations. For the Western blotting and immunostaining of Lyn-FG, its expression levels were enhanced by incubating the Lyn-FG–expressing HeLa Tet-On cells in medium supplemented with 2 µg/ml Dox for 24 h before the subsequent experiments.

### Fluorescence labeling and cross-linking of CD59, GM1, and DNP-DOPE
The anti-CD59 Ab IgG was purified from the supernatant of the culture medium of the mouse hybridoma MEM43/5 cell line (provided by V. Horejsi; Stefanová et al., 1991), and the anti-CD59 Fab was prepared by papain digestion of anti-CD59 IgG, followed by protein G column chromatography. The D/Ps of the A633 conjugates with anti-CD59 Fab, anti-CD59 IgG, anti-DNP IgG, and CTXB were 0.3, 0.6, 1.4, and 0.8, respectively.

To fluorescently visualize CD59 without cross-linking, the cells were incubated with 0.14 µg/ml anti-CD59 Fab-A633 in HBSS buffered with 2 mM Pipes at pH 7.4 (P-HBSS) at 27°C for 3 min. To generate CD59 clusters, the cells were first incubated with 0.5 µg/ml anti-CD59 IgG-A633 in P-HBSS at 27°C for 3 min and then with 1.8 µg/ml anti-mouse-IgG Abs produced in goat (ICN Biomedical) at 27°C for 10 min. To label GM1, cells were incubated with 1 nM CTXB-A633 in P-HBSS at 27°C for 2 min, which could cross-link up to five GM1 molecules. To generate larger GM1 clusters, after the GM1 labeling with CTXB-A633, the

CTXB-A633 was further cross-linked by the addition of goat anti-CTXB Abs (MilliporeSigma), diluted 1:100 with P-HBSS, at 27°C for 10 min.

DNP-DOPE was synthesized essentially as described previously (Murase et al., 2004). Briefly, after conjugating 2,4-dinitrophenyl-*N*-hydroxysuccinimide ester (Bayer Schering Pharma) to the amine group of DOPE (Avanti Polar Lipid), DNP-DOPE was purified by silica gel TLC and dissolved in methanol. For observing monomeric DNP-DOPE in the PM, the cells were first incubated with the direct addition of 1 µl of 1 mM DNP-DOPE (in methanol), and then the DNP-DOPE incorporated in the PM was labeled by incubating the cells in HBSS containing 5 nM A633–anti-DNP half-IgG and 1% BSA at 27°C for 3 min. To generate DNP-DOPE clusters in the PM, the cells in P-HBSS were first incubated with 1 µM DNP-DOPE at 27°C for 15 min, followed by incubation with 100 nM A633–anti-DNP IgG in P-HBSS containing 1% BSA at 27°C for 2 min, then with 170 nM goat anti-rabbit IgG (Cappel Laboratories) in the same buffer at 27°C for 15 min.

### Estimation of the cluster sizes of CD59, GM1, Lyn-FG, and FGH-Ras

The signal intensities of individual fluorescence spots representing one or more molecules on the PM were estimated by total internal reflection fluorescence microscopy, as described previously (Iino et al., 2001). Briefly, the fluorescence signal intensities of 600-nm × 600-nm areas (8-bit images in an area of 12 × 12 pixels), each containing a single spot, were measured. The background intensity estimated in adjacent areas was always subtracted. Histograms were fitted with a multipeak Gaussian by using Origin5 (OriginLab Corp.). In the case of CD59 clusters (Fig. 2 B, bottom), the histogram was fitted with the sum of five Gaussian functions, using the initial values for the means of $m$, $2m$, $3m$, $4m$, and $5m$, and those for the SDs of $\sigma$, $2^{1/2}\sigma$, $3^{1/2}\sigma$, $2\sigma$, and $5^{1/2}\sigma$, respectively, where $m$ and $\sigma$ are the mean signal intensity and SE for the spots representing single A633-Fab molecules adsorbed on the coverslip, with a certain range limitation for the value of each parameter. This provided a ratio of the five Gaussian integrated components of 18:31:31:18:2.

However, because the D/P of anti-CD59 IgG-A633 was 0.6; that is, 55% of CD59 molecules are not fluorescently labeled (according to the Poisson distribution), and this ratio does not represent the true distribution of the sizes of CD59 clusters (in terms of the number of IgG-A633 molecules in a cluster; e.g., a cluster of three CD59 proteins might exist without any fluorescence signal). From the Poisson distribution of a mean D/P of 0.6, the distributions of the molecules with true D/Ps of 0, 1, 2, and 3 are calculated to be 55:33:9.9:2.0, respectively. We simplified this ratio to 6:3:1:0, and, based on this distribution, the signal intensity distribution of real CD59 $N$-mers was calculated for the $n$ values of 3, 4, 5, and 6. When $n = 5$ (pentamers), the fractions of the fluorescent dye molecules in the fluorescence spots with the mean signal intensities of $m$, $2m$, $3m$, $4m$, and $5m$ became 13:40:30:16:1, respectively, which are closest to the observed ratio of 18:31:31:18:2. Therefore, although dimers, trimers, tetramers, hexamers, and so forth must exist, we believed that CD59 pentamers are the most frequent CD59 clusters. However,

in the present study, the fluorescent label was not on CD59 but on the anti-DC59 Ab IgG, and because the efficiency of divalent Ab binding to two CD59 molecules is probably very high due to the two-dimensionality of the CD59 spatial distribution on the PM (Grasberger et al., 1986), we concluded that the most frequently formed CD59 clusters consisted of 10 CD59 molecules.

In the case of Ab-CTXB-GM1 clusters (Fig. 7 B), the histogram could be fitted with a sum of four Gaussian functions, providing a ratio of 40:42:15:3 for the four integrated components. The D/P for A633-CTXB was 0.8. From the Poisson distribution of a mean D/P of 0.8, the distribution of the molecules with true D/Ps of 0, 1, 2, and 3 is calculated to be 45:36:14:4, respectively. We simplified this ratio to 5:4:1:0, and, based on this distribution, the signal intensity distribution of the $N$-mers of CTXB (Ab-CTXB-GM1 clusters) was calculated for the $n$ values of 1, 2, 3, and 4. When $n = 3$ (trimers of CTXB), the fractions of the fluorescent dye molecules in the fluorescence spots with the mean signal intensities of $m$, $2m$, $3m$, and $4m$ became 37:37:23:3, respectively, which are closest to the observed ratio of 40:42:15:3. As in the case with the CD59 clusters, although dimers, tetramers, pentamers, hexamers, and so forth must exist, we believe that the most frequent Ab-CTXB-GM1 clusters are those based on CTXB trimers. Using the same argument as for CD59 clusters, each CTXB is expected to be bound by 5 GM1 molecules, and thus we expect that each Ab-CTXB-GM1 cluster usually contains 15 GM1 molecules. This number is quite comparable to the presence of 10 CD59 molecules in a CD59 cluster raft.

### Cross-linking FGH-Ras and Lyn-FG on the cytoplasmic surface of the PM

AP20187, containing two binding sites for the FKBP protein, and AP21998, containing a single binding site for the FKBP protein (i.e., a control molecule for AP20187), were obtained from ARIAD Pharmaceuticals and stored in ethanol, according to the manufacturer's recommendations. The FKBP fusion proteins FGH-Ras and Lyn-FG, which contain two tandem FKBP moieties in a single molecule, were cross-linked by incubating the cells with 10 nM AP20187 in the culture medium at 27°C for 10 min.

### Detection of Erk phosphorylation after the induction of CD59 clusters and GM1 clusters

HeLa cells (30% confluence in a 60-mm dish) were cultured in MEM without serum for 36 h before the assay. The following incubations with Abs, CTXB, and EGF were performed in MEM. CD59 clusters were induced first by incubating the cells with 1.5 µg/ml anti-CD59 IgG at 37°C for 10 min and then with 1.8 µg/ml goat antimouse IgG at 37°C for 10 min. The control specimen without cross-linking was produced by incubating the cells with 1.5 µg/ml anti-CD59 Fab at 37°C for 10 min. To generate Ab-CTXB-GM1 clusters, first the cells were incubated with 18 nM CTXB at room temperature for 3 min, and then the CTXB-5GM1 was further cross-linked by adding goat anti-CTXB Abs (diluted 1:100 with MEM) at 37°C for 10 min. To produce positive control specimens, the cells were incubated with 20 nM EGF at 37°C for 5 min. For Western blot analyses, the cells were extracted on ice for 10 min with 0.3 ml ice-cold extraction buffer containing 1% NP-40, 0.25% sodium deoxycholate, 150 mM NaCl, 1 mM EDTA,

0.1% protease inhibitor mix (MilliporeSigma), and phosphatase inhibitors (1 mM $Na_2VO_3$ and 1 mM NaF), buffered with 50 mM Tris-HCl at pH 7.4. After brief centrifugation (15,000 rpm for 10 min), the supernatant was mixed with 0.1 ml of 4× sample buffer and incubated at 95°C for 5 min. The proteins in the extract were separated by SDS-PAGE, and then Western blotting was performed using rabbit anti-pErk1/2 (phospho-p44/42 MAP kinase [Thr202/Tyr204] Abs; Cell Signaling Technology).

### Evaluating the activation (biological function) of FGH-Ras by cross-linking with AP20187 or the addition of EGF

GST-RBD was expressed in *Escherichia coli* and purified with glutathione-Sepharose beads (Amersham). HeLa Tet-Off cells expressing FGH-Ras (30% confluence in a 10-cm dish) were cultured without serum 24–48 h before the assay. After treating the cells with 10 nM AP20187 or AP21998 for 10 min or 20 nM EGF for 5 min at 37°C, the cells were extracted on ice for 10 min with 1 ml ice-cold buffer containing 120 mM NaCl, 10% glycerol, 0.5% Triton X-100, 10 mM $MgCl_2$, 2 mM EDTA, 1 µg/ml aprotinin, and 1 µg/ml leupeptin buffered with 20 mM Hepes at pH 7.5 (assay buffer). After brief centrifugation (15,000 rpm for 10 min), 20 µl of an RBD-GST/glutathione-Sepharose bead suspension, prepared as described previously (de Rooij and Bos, 1997; Sydor et al., 1998), was added to the supernatant, and the mixture was incubated at 4°C for 1 h. The activated H-Ras molecules would become bound to the RBD-GST–conjugated beads in this process. The beads were then precipitated by centrifugation at 15,000 rpm for 1 min, washed three times with assay buffer, and then, after the final centrifugation, the pellet was mixed with 50 µl SDS sample buffer. After SDS-PAGE, Western blotting was conducted with mouse anti-Ras Abs (BD Transduction Laboratories).

### Evaluating the activation (biological function) of Lyn-FG by cross-linking with AP20187 or by antigen stimulation using RBL cells

RBL-2H3 cells expressing Lyn-FG (30% confluence in a 10-cm dish) were cultured without serum 24–48 h before the assay. The high-affinity Fcε receptor was bound by anti-DNP IgE by incubating the cells with 1 µg/ml anti-DNP IgE (MilliporeSigma) overnight. The cells were then incubated with 10 nM AP20187 or AP21998 for 10 min or 100 ng/ml DNP-BSA (MilliporeSigma) at 37°C for 60 min, and then extracted on ice for 10 min with 0.3 ml ice-cold extraction buffer containing 1% NP-40, 0.25% sodium deoxycholate, 150 mM NaCl, 1 mM EDTA, 0.1% protease inhibitor mix (MilliporeSigma), and phosphatase inhibitors (1 mM $Na_2VO_3$ and 1 mM NaF) buffered with 50 mM Tris-HCl at pH 7.4. After brief centrifugation (15,000 rpm for 10 min), the supernatant was mixed with 0.1 ml 4× sample buffer and incubated at 95°C for 5 min. The proteins in the extract were separated by SDS-PAGE, and then Western blotting was performed with rabbit anti-pY418 Abs (BioSource International) and rabbit anti-Lyn Abs (Santa Cruz Biotechnology).

### Online supplemental material

Fig. S1 shows that <10% of CD59 cluster rafts and Ab-CTXB-GM1 clusters became trapped in caveolae within 10 min after their induction. Fig. S2 shows that small clusters of FGH-Ras and Lyn-FG formed in the inner leaflet triggered signal transduction. Fig. S3 displays single-step photobleaching of a GFP monomer adsorbed on the glass and a Lyn-FG molecule in the PM observed at 200 Hz. Table S1, Table S2, and Table S3 summarize the colocalization lifetimes ($τ_1$, $τ_2$) and statistical parameters for molecular recruitment. Video 1 shows typical single-molecule image sequences of a transient colocalization event of a CD59 cluster and a single Lyn-FG molecule recorded at a 6.45-ms resolution (155 Hz).

## Acknowledgments

We thank Prof. A. Yoshimura for kind gifts of the cDNAs encoding GFP-H-Ras and GST-RBD, Prof. T. Fujimoto for the cDNA encoding cavelin-1–GFP, Prof. V. Horejsi of the Academy of Sciences of the Czech Republic (Staré Město, Czech Republic) for the mouse hybridoma MEM43/5 cell line, and Prof. N. Mizushima of the University of Tokyo School of Medicine for eager encouragement.

This work was supported in part by grants-in-aid for scientific research from the Japan Society for the Promotion of Science (Kiban C to I. Koyama-Honda [JP17K07302], Kiban B to T.K. Fujiwara [16H04775], Kiban C to R.S. Kasai [17K07333], Kiban B to K.G.N. Suzuki [18H02401], and Kiban S to A. Kusumi [16H06386]). The Institute for Integrated Cell-Material Sciences of Kyoto University is supported by the World Premiere Research Center Initiative of the Ministry of Education, Culture, Sports, Science and Technology of Japan.

The authors declare no competing financial interests.

Author contributions: I. Koyama-Honda performed a large majority of the single fluorescent molecule–tracking experiments. I. Koyama-Honda, E. Kajikawa, and H. Tsuboi conducted biochemical and cell biological experiments. I. Koyama-Honda, T.K. Fujiwara, R.S. Kasai, and A. Kusumi developed and built the single-molecule imaging station that can operate at higher frequencies. T.K. Fujiwara, R.S. Kasai, and T.A. Tsunoyama developed the analysis software. I. Koyama-Honda, K.G.N. Suzuki, and A. Kusumi conceived and formulated the project. I. Koyama-Honda, T.K. Fujiwara, R.S. Kasai, K.G.N. Suzuki, and A. Kusumi evaluated the obtained data and extensively discussed experimental plans during the entire course of this research. I. Koyama-Honda and A. Kusumi wrote the manuscript, and all authors participated in revisions.

Submitted: 21 June 2020

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

# Supplemental material

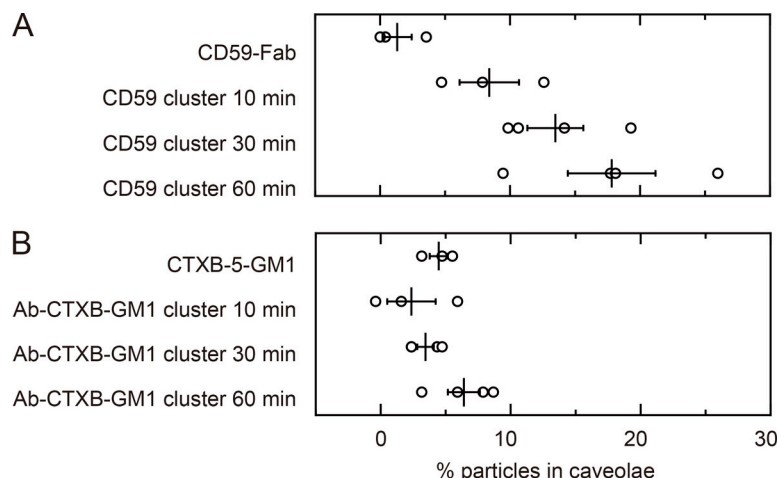

Figure S1. **Less than 10% of CD59 cluster rafts and Ab-CTXB-GM1 clusters became trapped in caveolae within 10 min after their induction. (A and B)** The fractions of CD59 cluster rafts (A) and Ab-CTXB-GM1 clusters (B) colocalized with the caveolin-1–GFP spots, as detected by simultaneous, two-color imaging at single-molecule sensitivity. The colocalized fractions increased with time but remained at <10% within 10 min after the cluster formation initiation. Based on these results, all of the experiments for observing the recruitment of intracellular molecules at CD59 cluster rafts and Ab-CTXB-GM1 clusters were performed within 10 min after cluster induction.

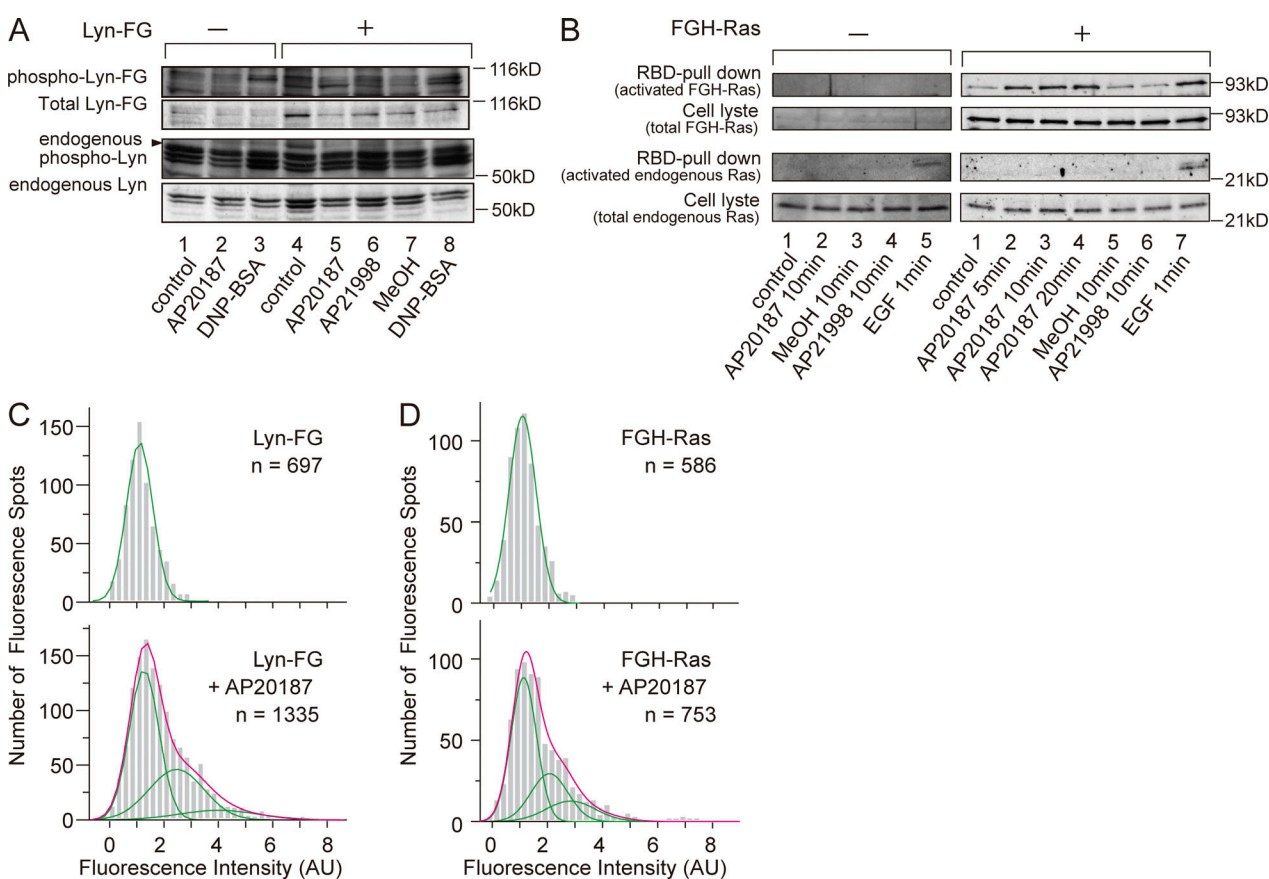

Figure S2. **Small clusters of FGH-Ras and Lyn-FG formed in the inner leaflet triggered signal transduction. (A)** Lyn-FG is likely functional because it exhibited self-phosphorylation (activation) similar to endogenous Lyn in RBL-2H3 cells after stimulation using anti-DNP IgE + DNP-BSA. Meanwhile, Lyn-FG oligomers induced by AP20187, an FKBP cross-linker, failed to activate Lyn-FG. Phosphorylation was detected by using the anti-pY418 Abs (taking the ratio of the anti-pY418 band versus the anti-Lyn band). **(B)** FGH-Ras is likely functional because it was pulled down, like endogenous Ras, by the Ras-binding domain (RBD) of the downstream molecule c-Raf kinase bound to polystyrene beads (detection with anti-Ras Abs) after EGF stimulation. FGH-Ras oligomerization by the addition of AP20187 induced FGH-Ras activation. AP21998 and methanol (MeOH) are negative controls. **(C and D)** Histograms showing the distributions of the signal intensities of individual fluorescence spots of Ly-FG (C) and FGH-Ras (D) before (top) and after (bottom) the addition of AP20187. Based on these histograms, we concluded that each Lyn-FG cluster and FGH-Ras cluster contained an average of approximately three Lyn-FG and FGH-Ras molecules, respectively.

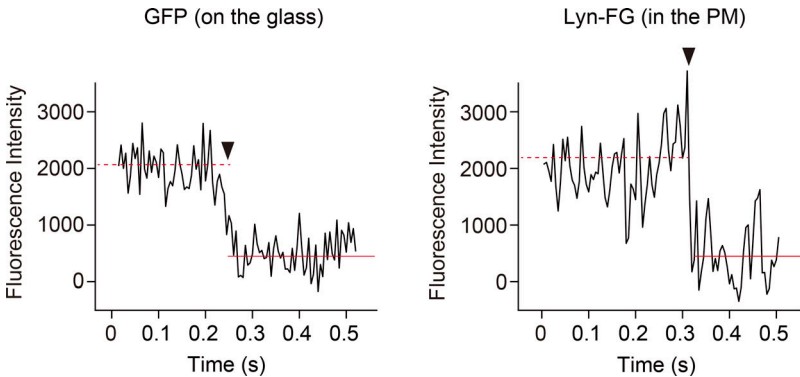

Figure S3. **Single-step photobleaching of a GFP monomer adsorbed on the glass and an Lyn-FG molecule in the PM observed at 200 Hz.** Typical time-dependent signal intensity changes of the fluorescence spots (observed in 15 × 15–pixel areas) of a recombinant GFP molecule sparsely adsorbed on the coverglass (left) and an Lyn-FG molecule in the HeLa cell PM (right). A large stepwise decrease in each panel represents a single-step photobleaching of the fluorescence spot (arrowhead), indicating that virtually every fluorescence spot represented a single molecule. The signal level before photobleaching is shown by a red dashed line, whereas the background signal intensity is shown by a horizontal red solid line.

Video 1.   **Transient recruitment of a single Lyn-FG molecule in/on the inner leaflet at a CD59 cluster located in/on the outer leaflet.** Typical single-molecule image sequences obtained at a 6.45-ms resolution (155 Hz), showing a transient colocalization event of a CD59 cluster (magenta spots) and a single Lyn-FG molecule (green spots). Colocalization is indicated by white arrows. The video is played at 1 frame/s, 155× slowed from real time. The total number of frames is 20 (130 ms).

**Three tables are provided online, each of which summarizes the colocalization lifetimes ($\tau_1$, $\tau_2$) and statistical parameters for molecular recruitment. Table S1 summarizes the data related to recruitment of cytoplasmic lipid-anchored molecules at CD59 clusters located in the outer leaflet. Table S2 summarizes the data related to recruitment of cytoplasmic lipid-anchored molecules at Ab-CTXB-GM1 clusters located in the outer leaflet, as compared with results at CTXB-5-GM1 and DNP-DOPE clusters. Table S3 summarizes colocalization lifetimes ($\tau_1$, $\tau_2$) and statistical parameters for the recruitment of the outer-leaflet molecules CD59 and CTXB-5-GM1, at the artificially induced oligomers of cytoplasmic lipid-anchored signaling molecules in the inner leaflet.**

