## [Peer Review File · The Journal of Cell Biology]

High-speed single-molecule imaging reveals signal transduction by induced transbilayer raft phases

Ikuko Koyama-Honda, Takahiro Fujiwara, Rinshi Kasai, Kenichi Suzuki, Eriko Kajikawa, Hisae Tsuboi, Taka Tsunoyama, and Akihiro Kusumi

Corresponding Author(s): Akihiro Kusumi, Okinawa Institute of Science and Technology

Review Timeline:	Submission Date:	2020-06-21
	Editorial Decision:	2020-07-27
	Revision Received:	2020-08-09
	Editorial Decision:	2020-09-01
	Revision Received:	2020-09-08

Monitoring Editor: Ira Mellman

Scientific Editor: Melina Casadio

Transaction Report:

DOI: <https://doi.org/10.1083/jcb.202006125>

July 27, 2020

Re: JCB manuscript #202006125

Prof. Akihiro Kusumi
Okinawa Institute of Science and Technology
Membrane Cooperativity Unit
Onna-son
Okinawa 904-0495
Japan

Dear Prof. Kusumi,

Thank you for submitting your manuscript entitled "High-speed single-molecule imaging reveals signal transduction by induced transbilayer raft phases". Thank you for your patience with the peer review process; we are sorry for the delay in sending our decision to you. The manuscript was assessed by expert reviewers, whose comments are appended to this letter. We invite you to submit a revision if you can address the reviewers' key concerns, as outlined here.

You will see that the reviewers found the work of high-quality, elegant, and rigorous. Rev#2 stressed that these analyses provide important new documentation of the dynamics of raft-signaling processes. We agree that the work is a technical achievement offering descriptive but important data that goes farther than any in the last 25 years to illustrate the principle of bilayer coupling. The reviewers however raised valid points that require your attention in revision. Please respond directly to Reviewer #1's technical points (including, but not limited to, the temperature issue, also raised by Rev #3). Editorially, we are most interested in responses and/or improvements as per Rev #1's points #2 and #3. Their point #4 is also a valid one, rebutted by Reviewer #2; this issue should at least be acknowledged in the text.

Please let us know if you would like to discuss the revisions further or anticipate any issues addressing the referee feedback. We are happy to discuss as needed.

GENERAL GUIDELINES:

Text limits: Character count for an Article is < 40,000, not including spaces. Count includes title page, abstract, introduction, results, discussion, acknowledgments, and figure legends. Count does not include materials and methods, references, tables, or supplemental legends.

Figures: Articles may have up to 10 main text figures. Figures must be prepared according to the policies outlined in our Instructions to Authors, under Data Presentation, <http://jcb.rupress.org/site/misc/ifora.xhtml>. All figures in accepted manuscripts will be screened prior to publication.

IMPORTANT: It is JCB policy that if requested, original data images must be made available. Failure to provide original images upon request will result in unavoidable delays in publication. Please ensure that you have access to all original microscopy and blot data images before submitting your revision.

Supplemental information: There are strict limits on the allowable amount of supplemental data. Articles may have up to 5 supplemental figures. Up to 10 supplemental videos or flash animations are allowed. A summary of all supplemental material should appear at the end of the Materials and methods section.

As you may know, the typical timeframe for revisions is three to four months. However, we at JCB realize that the implementation of social distancing and shelter in place measures that limit spread of COVID-19 also pose challenges to scientific researchers. Lab closures especially are preventing scientists from conducting experiments to further their research. Therefore, JCB has waived the revision time limit. We recommend that you reach out to the editors once your lab has reopened to decide on an appropriate time frame for resubmission. Please note that papers are generally considered through only one revision cycle, so any revised manuscript will likely be either accepted or rejected.

Thank you for this interesting contribution to the Journal of Cell Biology. You can contact us at the journal office with any questions, cellbio@rockefeller.edu or call (212) 327-8588.

Sincerely,

Ira Mellman, Ph.D.
Editor, Journal of Cell Biology

Melina Casadio, Ph.D.
Senior Scientific Editor, Journal of Cell Biology

Reviewer #1 (Comments to the Authors (Required)):

This manuscript by Koyama-Honda et al. presents an optical imaging study of how "raftophilic" transmembrane proteins form a signaling platform at the plasma membrane. How proteins organize themselves to activate a signaling pathway on live cells is an important question in both basic biology and disease pathology. However, it has been challenging to answer this question due to the lack of tools with sufficient time and spatial resolutions. In this work, the authors increased the time resolution (i.e. increased the frame rate to 200 Hz) by simply reducing the region of interest (ROI) of EMCCD and increased the spatial resolution by analyzing the kinetics of protein cluster images under the principle of single-molecule localization. Additionally, they lowered the temperature of cells from 37 to 27 degrees C to slow down the kinetic process on the PM. Under these experimental conditions, the authors determined that "CD59-cluster rafts" that were formed

artificially (antibody-induced clustering of target proteins) recruit downstream signaling proteins such as Lyn and H-Ras transiently (< 0.1 s) and correlated these transient events with signaling activation. Determination of time constant of an event is important to understand the overall molecular kinetics that describes signaling activation in the PM. However, I have concerns about the design of the experiments and interpretation of the data as listed below and thus suspect that the short dwelling (< 0.1 s) of downstream proteins in the "CD59-cluster rafts" really occurs even in the artificial experimental system. Plus, the manuscript is not clearly written to provide sufficient reasoning behind their hypotheses, background, and why certain methods were chosen. For these reasons, I cannot recommend this manuscript for publication in JCB in its present form.

1. A "raft" formation is known to be mediated by protein-lipid interactions that are temperature sensitive. Thus lowering the temperature to 27 degrees C may have allowed detecting transient (< 0.1 s) protein recruitment events but must have changed the lipid biophysics and thus protein-lipid interactions. The effects of the lowered temperature condition on the molecular dynamics occurring at the PM and on the interpretation of the data should be discussed.

2. The main analysis scheme presented is to use a bi-exponential function to fit the histogram of the duration of colocalization of a "raft" and a downstream protein. However, the fitting quality is poor and it is not convincing that bi-exponential is a correct functional form to use.

3. The actual image data of artificial "CD59-cluster rafts (magenta)" and "single" Lyn molecules (green) in Fig. 4 and video 1 are confusing. Images were collected at an acquisition rate of 155 Hz. However, in this condition, it would not be possible to observe such bright single-molecule fluorophore images (Lyn). It is very likely that the observed Lyn-FG molecules are clusters of molecules. If this is the case, the analysis and interpretation should be significantly modified.

4. The authors assumed that artificial clustering of CD59 or GM1 using antibodies results in "raft" formation. This is a bold assumption and needs to be verified.

5. Overall, the authors need to discuss how their result using artificial experimental conditions (27 degrees C, antibody-induced protein clustering, etc) can be translated into the events occurring in intact, normal cells in physiological conditions.

There are other minor comments that the authors may consider to improve the manuscript.

1. How were the large spots of proteins localized for single-molecule tracking analysis? These spots do not look circular and symmetric and determination of the actual location would be very challenging.

2. Have the authors considered that the inner-leaflet protein recruitment to the outer-leaflet raft can depend on the size of the raft? The size of the "raft" could affect the number of inner-protein recruitment events.

3. The biochemical data in Fig 3 were obtained from cells at 27 degrees C? This is an important detail but is missing.

4. The estimation of the number of CD59 molecules in a raft using the multi-Gaussian fitting of the histogram is not convincing.

5. mBCD induces gross changes of the PM. The authors should consider using other conditions.

Reviewer #2 (Comments to the Authors (Required)):

This paper is a continuation of Aki Kusumi's pioneering studies on single-molecule diffusion of lipids and proteins on the plasma membrane of living cells. Previously, he has analyzed how lipid probes partition into raft domains, employing a time resolution of 33ms. Now he has improved this resolution 6-7 fold to about 5ms. This is painstaking and meticulous work, unusual these days when detail does not get the attention that research of this type deserves. The authors have used lipid-anchored probes including Lyn and H-R as well as a ganglioside GM1 partitioning into the outer leaflet of lipid raft clusters and DOPE as a non-raft phospholipid. For me it is important to stress how carefully the Kusumi group has developed its set of probes by synthesizing these fluorescent molecules themselves, testifying to their attention to detail that decides over success and failure. The GM1 ganglioside is the only ganglioside that indeed has all the properties required to mimic the natural non-fluorescent molecule so far.

Koyama et al proceeded to analyze how CD59 clusters and GM1 clusters trigger the activation of Lyn and H-Ras by recruiting them to the cytoplasmic leaflet of the raft clusters. Detailed studies of the diffusion paths and kinetics showed how these signalling molecules became bound and released from the raft clusters during the signal activation process. To me this study is a model example of the obsessively high quality of Kusumi's work. Someone could of course complain and say that they have in principle repeated experiments done before with similar results. Or that the study uses antibodies or cholera toxin to activate signalling. This is not the real thing because the clustering is artefactual. But this type of comment misses the point. Almost everything we do has some artefactual elements embedded in the methodology. Both clustering modes activate signaling. Thus the study analyzes signaling under well defined conditions. In a field which has been plagued by so much controversy, studies of this quality must be published. Studies of this quality would be a bonus for any field. The authors demonstrate the dynamics of raft-signalling processes not documented before. I have nothing to add but my recommendation that this paper should be published by JCB with high priority.

Reviewer #3 (Comments to the Authors (Required)):

The paper by Koyana-Honda et al. uses single molecule imaging at ultra-high speeds to describe the clustering characteristics of signaling molecules (i.e., Lyn, H-Ras and ERK) on the inner PM bilayer in response to clustering of GPI-anchored CD59 on the outer leaflet. Their results show CD59 clustering triggers transient recruitment and activation of the inner leaflet signaling molecules. This supports the idea of transbilayer coupling, in which raft phases induced on one leaflet of the PM (i.e., through crosslinking of CD59) lead to the formation of a raft phase on the inner leaflet that attracts raft-associating signaling molecules (i.e., Lyn, H-Ras and ERK). As is typical of the Kusumi lab, the work is elegantly performed, rigorous and uses state-of-the-art imaging technologies, including a ~5 fold improvement in temporal resolution of signaling molecules. While the Jitu Mayor lab has used homo-FRET approaches to propose a transbilayer coupling mechanism, Koyana-Honda et al. here are the first to show transbilayer coupling at the single

molecule level in response to antibody cross-linking. This makes the study appropriate for publication in JCB. The study could be improved by addressing a few additional points, as described below.

1. The authors performed all their experiments at 27°C. As phase-partitioning in membranes is very temperature sensitive, being induced at low temperatures, can the authors demonstrate that they can see transbilayer coupling at more physiological temperatures?
2. One problem the authors face is the short recording periods due to photobleaching of dye fluorescence on their antibodies. Have the authors considered using more photobleaching-resistant Halo-dyes to perform their studies? This might allow longer recording periods.
3. On page 9, third sentence from the section on Lyn, the authors state that virtually all Lyn-FG molecules undergo thermal diffusion and cite Fig. 1A as evidence. Figure 1A is a diagram so this must be incorrect. The authors need to correctly point to the proper figure for this.
4. On page 20, second line, the authors mention 'the order of 10s of minutes'. It is not clear what this means- is it tens of minutes or tens of seconds?
5. The authors need to extend their discussion related to whether the signaling molecules that show transient localization at cross-linked CD59 sites are doing so through a mechanism of lipid phase partitioning alone, by actual transbilayer coupling (through interactions across the bilayer), or both.

August 9, 2020

Ira Mellman, Ph.D.
Editor, Journal of Cell Biology

Melina Casadio, Ph.D.
Senior Scientific Editor, Journal of Cell Biology

Re: JCB manuscript #202006125, entitled
"High-speed single-molecule imaging reveals signal transduction by induced transbilayer raft phases"

Dear Ira and Melina,

Thank you very much for critically reading and assessing our manuscript. We would also like to thank you for obtaining the opinions of the three referees. Attached please find our revised manuscript.

We have addressed all of the points raised by you and your referees in the revised manuscript. All of the recommendations have been basically complied with. In particular, the concerns emphasized by you, i.e., the Reviewers #1 and #3's point #1 as well as the Reviewer #1's point #2, 3, and 4, were carefully considered and addressed in the revised manuscript.

As a result, we believe that the manuscript has been considerably strengthened. We would like to thank you and your reviewers again for critically reading our manuscript and providing constructive comments and recommendations. We hope this manuscript is now acceptable for publication in *The Journal of Cell Biology*.

The revisions made in the text are indicated by blue highlighting. Our point-by-point responses to your reviewers' comments are provided on the following pages.

Sincerely yours,

Aki (Akihiro Kusumi)
Professor
Membrane Cooperativity Unit
Okinawa Institute of Science and Technology Graduate University (OIST)
e-mail: akihiro.kusumi@oist.jp

Reviewer #1

This manuscript by Koyama-Honda et al. presents an optical imaging study of how "raftophilic" transmembrane proteins form a signaling platform at the plasma membrane. How proteins organize themselves to activate a signaling pathway on live cells is an important question in both basic biology and disease pathology. However, it has been challenging to answer this question due to the lack of tools with sufficient time and spatial resolutions. In this work, the authors increased the time resolution (i.e. increased the frame rate to 200 Hz) by simply reducing the region of interest (ROI) of EMCCD and increased the spatial resolution by analyzing the kinetics of protein cluster images under the principle of single-molecule localization. Additionally, they lowered the temperature of cells from 37 to 27 degrees C to slow down the kinetic process on the PM. Under these experimental conditions, the authors determined that "CD59-cluster rafts" that were formed artificially (antibody-induced clustering of target proteins) recruit downstream signaling proteins such as Lyn and H-Ras transiently (< 0.1 s) and correlated these transient events with signaling activation. Determination of time constant of an event is important to understand the overall molecular kinetics that describes signaling activation in the PM. However, I have concerns about the design of the experiments and interpretation of the data as listed below and thus suspect that the short dwelling (< 0.1 s) of downstream proteins in the "CD59-cluster rafts" really occurs even in the artificial experimental system. Plus, the manuscript is not clearly written to provide sufficient reasoning behind their hypotheses, background, and why certain methods were chosen. For these reasons, I cannot recommend this manuscript for publication in JCB in its present form.

Thank you very much for your critical reading of our manuscript. We addressed all of the points raised by you and revised the manuscript accordingly.

1. A "raft" formation is known to be mediated by protein-lipid interactions that are temperature sensitive. Thus lowering the temperature to 27 degrees C may have allowed detecting transient (< 0.1 s) protein recruitment events but must have changed the lipid biophysics and thus protein-lipid interactions. The effects of the lowered temperature condition on the molecular dynamics occurring at the PM and on the interpretation of the data should be discussed.

As pointed out by Reviewer 1, raft formation is temperature dependent. This is most pronounced in many cell lines, when actin-based membrane skeleton meshes are removed from the PM cytoplasmic surface, at temperatures below $\sim 15^{\circ}\text{C}$, at which large raft domains are induced and become visible by fluorescence microscopy, by the process resembling the phase transition: This would not occur at 27°C (Holowka et al., 1983; Gidwani et al., 2001; Veatch and Keller, 2003; Baumgart et al., 2007; Lingwood et al., 2008; Sengupta et al., 2008; Levental et al., 2009; Kusumi et al., 2020). Namely, the changes in the lipid biophysics and protein-lipid interactions that occur when the temperature is lowered from 37°C to 27°C would be rather quantitative than qualitative. For example, the diffusion coefficients of various lipids and GPI-ARs in two very different cell types, CHO and RBL-2H3 cells, at 27°C were reported to decrease only by a factor of at most 1.4 when the temperature was lowered from 37°C to 27°C (Lee et al., 2015; Sal et al., 2015). Meanwhile, the diffusion coefficients of both the prototypical non-raft phospholipid, L- α -dioleoylphosphatidylcholine, and the prototypical raft-associated phospholipids, C18-sphingomyelin and L- α -distearoylphosphatidylcholine (all of

them were fluorescently labeled), would be reduced by a factor of approximately two when the temperature was lowered from 37°C to 27°C (assuming that the activation energy for diffusion is the same between 37°C and 23°C) (Kinoshita et al., 2017, where we used T24 and PtK2 cells; based on these results, we decided to perform all of the microscope observations at 27°C to better detect the colocalization processes). Therefore, we believe that the conclusions obtained in the present work based on the observations performed at 27°C are essentially correct. These descriptions have now been added **on p. 9** in the revised manuscript.

2. The main analysis scheme presented is to use a bi-exponential function to fit the histogram of the duration of colocalization of a "raft" and a downstream protein. However, the fitting quality is poor and it is not convincing that bi-exponential is a correct functional form to use.

The Brownian simulation and theory predicted that, in the absence of binding (under the conditions where only incidental colocalizations take place), the distribution of the colocalization durations could be approximated by a single exponential function, at the present time resolution (for example, see Sungkaworn et al., 2017). At higher time resolutions, we expect to observe the sum of exponential functions (infinite numbers; the time constants become shorter with an increase of n), but these faster decays were not observable under our experimental setting. So, we assumed that the incidental colocalization lifetime is described by a single exponential function, $\exp(-t/\tau_1)$. As long as this assumption holds, the final functional form (the addition of two exponential functions, as described in the original manuscript) should approximate the experimental histograms quite well.

In the original manuscript, we simply stated, "histogram $h(\text{incidental-by-shift})$, which is proportional to $\exp(-t/\tau_1)$ (t ; time) at the present experimental accuracies" in the section of "**Colocalization detection and evaluation of colocalization lifetimes**" in the **Materials and methods**, because virtually all of the single-molecule studies measuring the binding or colocalization lifetimes assumed this (for example, Sungkaworn et al., [2017], as cited in the previous paragraph). We agree that this clause was too simple and so we added the explanation described in the previous paragraph in the revised manuscript (**p. 30**).

We agree that the fitting was, in some cases, not as good as we hoped to see, despite quite tremendous efforts we made to improve the signal-to-noise ratio of the instruments. However, we performed extensive statistical tests and provided SEMs for the determined values. Therefore, although the fitting was sometimes poor, the conclusions derived from these analyses must be correct. For the actual binding time, $\tau_B (= \tau_2)$, we explicitly stated, in the original manuscript, that we would pay more attention to the presence of two components in the colocalization duration histograms, rather than the actual values of τ_2 in **Results on p. 12** and in **Materials and methods** [now in the **last paragraph on p. 29** and the **first paragraph on p. 30**]). To further clarify this point, we added the following phrase on **p. 12**, "due to large errors involved in the determinations of τ_2 ".

3. The actual image data of artificial "CD59-cluster rafts (magenta)" and "single" Lyn molecules (green) in Fig. 4 and video 1 are confusing. Images were collected at an acquisition rate of 155 Hz. However, in this condition, it would not be possible to observe such bright single-molecule fluorophore images (Lyn). It is very likely that the observed Lyn-FG molecules are clusters of molecules. If this is the case, the analysis and interpretation should be significantly modified.

Compared with our previous observation conditions and instrument (Koyama-Honda et al., 2005), the optical components of our single-molecule imaging station have been improved, image intensifiers were new (the same product, but used anew and so the photocathodes were in better conditions), and the excitation laser intensities have been increased by a factor of about two. Therefore, despite the increase in the frame rate, single-molecule localization precisions remained about the same as before. So, the fluorescent spots of Lyn-FG (and other single molecules) were quite bright, as noted by Reviewer 1.

4. The authors assumed that artificial clustering of CD59 or GM1 using antibodies results in "raft" formation. This is a bold assumption and needs to be verified.

As described in **Introduction**, we have already shown extensively in our previous publications that artificial CD59 clusters form signaling raft domains (Suzuki et al., 2007a,b, JCB; Suzuki et al., 2012, Nat. Chem. Biol.). Raft-like properties of artificial CD59 clusters were also shown by us by the finding that gangliosides and sphingomyelins are colocalized with the artificial CD59 clusters (Komura et al., 2016, Nat. Chem. Biol.; Kinoshita et al., 2017, JCB). CD59-TM, in which the GPI-anchor was replaced by the transmembrane domain of a prototypical non-raft molecule, LDL receptor, failed to exhibit the raft-like behaviors and to trigger the downstream signal, in ways similar to the CD59 clusters after cholesterol depletion (Suzuki et al., 2007a,b, JCB; Suzuki et al., 2012, Nat. Chem. Biol.). The present research was designed based on these previous research results.

Due to the length limit of the journal, we described these previous results quite concisely in our original manuscript, and so our explanations might not have been sufficient. So, we extensively revised this part (please see **pp. 4-5**). Furthermore, we added the description of our previous observations in which artificially-induced CD59 signal was quite similar to the signal triggered by the complement component C8 or the membrane attack complement complexes (Suzuki et al., 2007a,b, JCB; Suzuki et al., 2012, Nat. Chem. Biol.) (**p. 4, lines 6-9**).

With regard to GM1 clusters, the stabilization and enlargement of raft domains induced by CTXB and its antibodies as well as the signaling triggered by the enhanced raft domains have been established quite well in the literature, although the actual data shown there were quite qualitative (reviewed by Kusumi et al., 2020). To further explain this point in the revised manuscript, we explained the result of a prominent study by Janes et al. (1999) using the T-cell line, E6.1 Jurkat (This report was cited in a different context in the original manuscript). In this report, Janes et al. found that the addition of CTXB and its antibody induced membrane patches containing LCK, LAT, and the T-cell receptor (TCR), but excluding CD45. These patches were considered to be enhanced raft domains because they were colocalized by CD59, used as a prototypical raft marker in the study by Janes et al. (1999). The present study is based on these previous observations. This is now summarized on **pp. 18-19** in the revised manuscript.

Also, please refer to the comment by Reviewer 2. "Someone could of course complain that the study uses antibodies or cholera toxin to activate signalling. This is not the real thing because the clustering is artefactual. But this type of comment misses the point. Almost everything we do has some artefactual elements embedded

in the methodology. Both clustering modes activate signaling. Thus the study analyzes signaling under well defined conditions.”

5. Overall, the authors need to discuss how their result using artificial experimental conditions (27 degrees C, antibody-induced protein clustering, etc) can be translated into the events occurring in intact, normal cells in physiological conditions.

The possible problems of employing artificial conditions of observing molecular interactions at 27°C rather than 37°C and the use of antibody-induced protein clustering are now discussed, respectively, on **p. 9** in **Results** and **p. 4** in **Introduction** in the revised manuscript (the same places of our responses to the Reviewer 1’s major points 1 and 4, respectively).

There are other minor comments that the authors may consider to improve the manuscript.

1. How were the large spots of proteins localized for single-molecule tracking analysis? These spots do not look circular and symmetric and determination of the actual location would be very challenging.

Due to the problems of lower signal-to-noise ratios and transient faster movements that sometimes occurred due to the stochastic nature of the thermal molecular movements, the single-molecule images are not always circular (two-dimensional Gaussian). This is a general problem of single-molecule tracking. To deal with these problems, we used a cross-correlation method developed by Gelles et al. (1988), employing an ideal two-dimensional Gaussian distribution pattern as a kernel. The positions of single molecules and single clusters were operationally defined as the center of mass of the cross-correlation functions. We failed to describe this method, and so, it is now included in **Materials and methods (p. 27)**.

2. Have the authors considered that the inner-leaflet protein recruitment to the outer-leaflet raft can depend on the size of the raft? The size of the "raft" could affect the number of inner-protein recruitment events.

We totally agree that the size of the raft domains formed in the outer leaflet could also be related to the efficiencies of recruiting inner-leaflet raftophilic signaling molecules. This was already pointed out in **Results** in the original manuscript, but we further clarified this point in the same paragraph (**p. 17**). In addition, this point has been included in **Discussion (pp. 22-23)**.

3. The biochemical data in Fig 3 were obtained from cells at 27 degrees C? This is an important detail but is missing.

Thank you very much for pointing this out. The biochemical data in Fig. 3 was indeed obtained at 37°C. This is now indicated in the main text in the revised manuscript (**p. 10**). The description about the detailed method for obtaining the biochemical data shown in Fig. 3 was missing in the original manuscript, and so, we added the method (in **Materials and methods on pp. 35**).

4. The estimation of the number of CD59 molecules in a raft using the multi-Gaussian fitting of the histogram is not convincing.

These were our best estimates, but as pointed out by Reviewer 1, the estimates are not quite accurate. So, we moderated the expressions in the main text (**pp. 9 and p. 16**).

5. mBCD induces gross changes of the PM. The authors should consider using other conditions.

As pointed out by Reviewer 1, the use of the M β CD treatments (4 mM, 37°C, 30 min) has been controversial. However, the involvement of raft domains was examined in a variety of methods in the present research, including the use of various lipid anchoring chains and the transmembrane domain of a prototypical non-raft molecule, LDL receptor, and a prototypical non-raft phospholipid DOPE. In the past, we employed the M β CD treatments together with other control experiments (using artificial TM mutants of GPI-anchored receptors, saponin treatment, cholesterol repletion after the M β CD treatment), and found that the treatment with 4 mM M β CD at 37°C for 30 min reproducibly gave the results consistent with those obtained by using other methods of testing the raft involvement. This is now explained in **the caption to Fig. 6**.

Reviewer #2:

This paper is a continuation of Aki Kusumi's pioneering studies on single-molecule diffusion of lipids and proteins on the plasma membrane of living cells. Previously, he has analyzed how lipid probes partition into raft domains, employing a time resolution of 33ms. Now he has improved this resolution 6-7 fold to about 5ms. This is painstaking and meticulous work, unusual these days when detail does not get the attention that research of this type deserves. The authors have used lipid-anchored probes including Lyn and H-R as well as a ganglioside GM1 partitioning into the outer leaflet of lipid raft clusters and DOPE as a non-raft phospholipid. For me it is important to stress how carefully the Kusumi group has developed its set of probes by synthesizing these fluorescent molecules themselves, testifying to their attention to detail that decides over success and failure. The GM1 ganglioside is the only ganglioside that indeed has all the properties required to mimic the natural non-fluorescent molecule so far.

Koyama et al proceeded to analyze how CD59 clusters and GM1 clusters trigger the activation of Lyn and H-Ras by recruiting them to the cytoplasmic leaflet of the raft clusters. Detailed studies of the diffusion paths and kinetics showed how these signalling molecules became bound and released from the raft clusters during the signal activation process.

To me this study is a model example of the obsessively high quality of Kusumi's work. Someone could of course complain and say that they have in principle repeated experiments done before with similar results. Or that the study uses antibodies or cholera toxin to activate signalling. This is not the real thing because the clustering is artefactual. But this type of comment misses the point. Almost everything we do has some artefactual elements embedded in the methodology. Both clustering modes activate signaling. Thus the

study analyzes signaling under well defined conditions. In a field which has been plagued by so much controversy, studies of this quality must be published. Studies of this quality would be a bonus for any field. The authors demonstrate the dynamics of raft-signalling processes not documented before. I have nothing to add but my recommendation that this paper should be published by JCB with high priority.

Thank you very much for your extremely kind words and understanding of our research purposes and strategies.

Reviewer #3

The paper by Koyana-Honda et al. uses single molecule imaging at ultra-high speeds to describe the clustering characteristics of signaling molecules (i.e., Lyn, H-Ras and ERK) on the inner PM bilayer in response to clustering of GPI-anchored CD59 on the outer leaflet. Their results show CD59 clustering triggers transient recruitment and activation of the inner leaflet signaling molecules. This supports the idea of transbilayer coupling, in which raft phases induced on one leaflet of the PM (i.e., through crosslinking of CD59) lead to the formation of a raft phase on the inner leaflet that attracts raft-associating signaling molecules (i.e., Lyn, H-Ras and ERK). As is typical of the Kusumi lab, the work is elegantly performed, rigorous and uses state-of-the-art imaging technologies, including a ~5 fold improvement in temporal resolution of signaling molecules. While the Jitu Mayor lab has used homo-FRET approaches to propose a transbilayer coupling mechanism, Koyana-Honda et al. here are the first to show transbilayer coupling at the single molecule level in response to antibody cross-linking. This makes the study appropriate for publication in JCB. The study could be improved by addressing a few additional points, as described below.

Thank you very much for your extremely kind words and for your critical reading of the manuscript.

1. The authors performed all their experiments at 27°C. As phase-partitioning in membranes is very temperature sensitive, being induced at low temperatures, can the authors demonstrate that they can see transbilayer coupling at more physiological temperatures?

Please see our response to the first point of Reviewer 1.

2. One problem the authors face is the short recording periods due to photobleaching of dye fluorescence on their antibodies. Have the authors considered using more photobleaching-resistant Halo-dyes to perform their studies? This might allow longer recording periods.

Thank you very much for your advice. Since the colocalization durations we encountered in this study were much shorter than the GFP photobleaching lifetime, and so, for this particular study, we would stick to the GFP data. However, for observing behaviors with longer characteristic times, we would follow your advice (see, for example, our paper by Tsunoyama et al. 2018 Nat. Chem. Biol.).

3. On page 9, third sentence from the section on Lyn, the authors state that virtually all Lyn-FG molecules

undergo thermal diffusion and cite Fig. 1A as evidence. Figure 1A is a diagram so this must be incorrect. The authors need to correctly point to the proper figure for this.

Corrected.

4. On page 20, second line, the authors mention 'the order of 10s of minutes'. It is not clear what this means- is it tens of minutes or tens of seconds?

We changed to “tens of minutes”, following the advice of Reviewer 3.

5. The authors need to extend their discussion related to whether the signaling molecules that show transient localization at cross-linked CD59 sites are doing so through a mechanism of lipid phase partitioning alone, by actual transbilayer coupling (through interactions across the bilayer), or both.

Indeed, the punchline was missing in the main text although it was described in the caption to **Fig. 10**. This happened due to the space limitations: we tried to avoid too much duplications between the main text and the caption to **Fig. 10**. We think both are involved, but we could not identify an as-yet unknown TM protein(s) x. Following the advice of Reviewer 3, we added a few lines in the main text in **Discussion**, describing the possible involvement of both the transbilayer raft phase and TM proteins (**p. 24**).

September 1, 2020

RE: JCB Manuscript #202006125R

Prof. Akihiro Kusumi
Okinawa Institute of Science and Technology
Membrane Cooperativity Unit
Onna-son
Okinawa 904-0495
Japan

Dear Prof. Kusumi,

Thank you for submitting your revised manuscript entitled "High-speed single-molecule imaging reveals signal transduction by induced transbilayer raft phases". While Reviewers #2 and #3 now recommend publication, you will see that Reviewer #1 has remaining concerns that require your attention. We appreciated that Reviewers #2-3 shared with us that they found your responses to the concerns about the effects of the temperature on the molecular dynamics at the PM and on the interpretation of the data satisfactory. Therefore, we would not require further experimentation to address Reviewer #1's point #1. However, please do respond to this reviewer's other remaining points in a letter and with clarifications in the text, where needed. Perhaps keeping the duration data in 6D, stating in the paper the caveats in using this function, and adding the panel that the ref suggests would help assuage their concerns (point #2). Discussion seems appropriate to tackle Reviewer #1's point #3. Overall, we would be happy to publish your paper in JCB pending final revisions necessary to meet our formatting guidelines (see details below) and pending appropriate responses to the remaining reviewer comments as outlined above.

1) eTOC summary: A 40-word summary that describes the context and significance of the findings for a general readership should be included on the title page. The statement should be written in the present tense and refer to the work in the third person.

****Please revise the eTOC statement to meet our preferred style: it should start with "First author name(s) et al..."****

2) Figure formatting:

Molecular weight or nucleic acid size markers must be included on all gel electrophoresis. Please add molecular weight with unit labels on the following panels: figure 3, S2AB

3) Statistical analysis: Error bars on graphic representations of numerical data must be clearly described in the figure legend. The number of independent data points (n) represented in a graph must be indicated in the legend. Statistical methods should be explained in full in the materials and methods. For figures presenting pooled data the statistical measure should be defined in the figure legends.

4) Materials and methods: Should be comprehensive and not simply reference a previous

publication for details on how an experiment was performed. Please provide full descriptions in the text for readers who may not have access to referenced manuscripts.

- Microscope image acquisition. The following information must be provided about the acquisition and processing of images:

a. Make and model of microscope

b. Type, magnification, and numerical aperture of the objective lenses

c. Temperature

d. imaging medium

e. Fluorochromes

f. Camera make and model

g. Acquisition software

h. Any software used for image processing subsequent to data acquisition. Please include details and types of operations involved (e.g., type of deconvolution, 3D reconstitutions, surface or volume rendering, gamma adjustments, etc.).

A. MANUSCRIPT ORGANIZATION AND FORMATTING:

Full guidelines are available on our Instructions for Authors page, <http://jcb.rupress.org/submission-guidelines#revised>. **Submission of a paper that does not conform to JCB guidelines will delay the acceptance of your manuscript.**

B. FINAL FILES:

-- High-resolution figure and video files: See our detailed guidelines for preparing your production-ready images, <http://jcb.rupress.org/fig-vid-guidelines>.

Thank you for your attention to these final processing requirements. Please revise and format the manuscript and upload materials within 7 days. If complications arising from measures taken to prevent the spread of COVID-19 will prevent you from meeting this deadline (e.g. if you cannot retrieve necessary files from your laboratory, etc.), please let us know and we can work with you to

determine a suitable revision period.

Thank you for this interesting contribution, we look forward to publishing your paper in the Journal of Cell Biology.

Sincerely,

Ira Mellman, Ph.D.
Editor, Journal of Cell Biology

Melina Casadio, Ph.D.
Senior Scientific Editor, Journal of Cell Biology

Reviewer #1 (Comments to the Authors (Required)):

I will mainly comment on the authors' responses to #1-4 of my past comments.

#1. I find it helpful that the authors addressed that actin-dependent phase transition of the plasma membrane (PM) generally occurs below 27°C as evidenced by various past studies. However, this manuscript is about measuring the kinetics of protein association that contributes to raft formation, a process that is known to control signal transduction. It is great that this lowered temperature allows the authors to acquire the kinetics data by enabling the visualizations of the raft formation and protein-protein interactions. However, if the temperature change affects the downstream signaling significantly without PM transitioning to a gel phase, the importance of the finding needs to be reassessed. Therefore, I strongly suggest the authors at least compare the biochemical readouts (WB) of signaling between 27°C and 37°C by repeating the same experiment shown in Fig 3 at 27°C. In my previous review, I thought this must have been done at 27°C because the biophysical measurements were all done at 27°C. Having this additional dataset will strengthen this manuscript and also assure other biophysists when they consider using 27°C for their own kinetics studies on the plasma membrane.

#2. I am still not convinced by this reasoning, "At higher time resolutions, we expect to observe the sum of exponential functions (infinite numbers; the time constants become shorter with an increase of n), but these faster decays were not observable under our experimental setting." because the number of components was arbitrarily set to 2 by the authors without clear justification. Since the same formula was used to analyze all the colocalization data and it is described that the authors only concern about the presence of the second component (they mentioned comparing tau2 values is meaningless due to the large errors) which implies the presence of protein-protein interaction, I suggest that the authors replace Fig 6D with another figure that only indicates the presence or absence of such interaction. Again, the current Fig. 6D is not helpful as the authors also mentioned that "As described in Materials and methods, tau2 directly represents the binding duration (inverse off-rate assuming simple zero-order dissociation kinetics for Lyn-FG from the CD59 cluster). However, note that, in the scope of this report, we paid more attention to the presence of two components in the colocalization duration histograms, rather than the actual values of tau2, due to large errors involved in the determinations of tau2."

3. While the authors claim that their improved detection conditions can enable to probe single Lyn-

FG molecules in solution, this is not convincing without actual data supporting such claim. Reference single molecule intensities, which can be estimated by anchoring dyes or fluorescent proteins to a glass substrate, are required to use them as baseline intensity values. It is also necessary that they discuss the possibilities that they were not looking at single molecules and if so, how that would affect the analysis and interpretation of the data.

4. I am happy with their responses.

I hope that the authors can address these concerns to improve their manuscript further for publication in JCB.

Reviewer #2 (Comments to the Authors (Required)):

Accept.

Reviewer #3 (Comments to the Authors (Required)):

The authors have addressed all my prior concerns. The paper is now ready for publication.

2nd Revision - Authors' Response to Reviewers: September 8, 2020

September 6, 2020

Ira Mellman, Ph.D.
Editor, Journal of Cell Biology

Melina Casadio, Ph.D.
Senior Scientific Editor, Journal of Cell Biology

Re: JCB manuscript #202006125R, entitled
"High-speed single-molecule imaging reveals signal transduction by induced transbilayer raft phases"

Dear Ira and Melina,

Thank you very much for critically reading and assessing our manuscript again, and for basically agreeing to publish our manuscript. Attached please find our second revised manuscript.

Following your advice about the points #2 and #3 raised by Reviewer 1, we have made the following changes. These changes are indicated by **yellow highlighting** in the present manuscript.

With regard to #2.

As suggested by you, we kept the duration data in **Fig. 6D**. Our justification for using two exponential functions has been written in Materials and methods, even in the original manuscript, and we thought that we further clarified it in the previously revised manuscript. However, since we now better understand the point raised by Reviewer 1, we further clarified the justification. Please see the parts **highlighted in yellow on pp. 29 - 30** (an addition of a new reference on **p. 43**). Since some statements we added in the previously revised manuscript (**p. 30**) now appear unrelated to the question first raised by Reviewer 1, they have been removed in the present manuscript. So, in this paragraph, the previously-made revision is indicated in **blue highlighting** and new additions are indicated by **yellow highlighting**.

The caveats you pointed out are now discussed on **p. 12** in the present manuscript (**Results**).

Reviewer 1 suggested that we conclude that protein-protein interaction is involved in the Lyn-FG recruitment to CD59 clusters. If we could, we would like to. However, as described in the manuscript, the colocalization lifetime of Lyn-FG at CD59 clusters was not statistically significantly longer than that of myrpal-N20(Lyn)-GFP at CD59 clusters although the measured lifetimes were 80 and 66 ms, respectively, i.e. the former was indeed longer than the latter (but with no statistical significance; $P = 0.068$; **Fig. 1A**; The colocalization lifetimes of FGH-Ras and GFP-tH at CD59 clusters were 91 and 75 ms, respectively, which were also non-significant). From these observations, it was impossible for us to conclude the importance of protein-protein interactions for the recruitment of Lyn-FG and FGH-Ras to CD59 clusters.

Therefore, we could not help but write our conclusion (one of our conclusions) in the following way: "that (3) when both the Lyn protein moiety and raftophilic myristoyl+palmitoyl chains exist, the lifetime at the

CD59 cluster raft appears to be prolonged (could be proven in the future when single-molecule imaging is further improved)" (green highlight on p. 14; no change from the original manuscript). Namely, although we hoped to make the statement suggested by Reviewer 1 (involvement of protein-protein interaction in the recruitment of Lyn-FG to CD59 clusters), already during the days of experiments and manuscript preparation, we would be unable to conclude the involvement of protein-protein interaction, until we could unequivocally show that the presence of the protein moiety prolongs the colocalization lifetime.

With regard to #3.

Rather than discussing this issue as you suggested, since we have had the data to show this (had done the experiments that Reviewer #1 suggested), we put them in **Supplementary Fig. S3** and on **p. 11** in the main text in the present manuscript.

Due to these revisions, we believe that the manuscript has been considerably strengthened. We would like to thank you and your reviewers again for critically reading our manuscript and providing constructive comments and recommendations.

In addition, we followed your instructions about the eTOC summary format, figure formatting, descriptions of n and the meaning of error bars, important details in Materials and methods, including software we used.

We hope that this manuscript is now formally accepted for publication in *The Journal of Cell Biology*.

Sincerely yours,

Aki (Akihiro Kusumi)
Professor
Membrane Cooperativity Unit
Okinawa Institute of Science and Technology Graduate University (OIST)
e-mail: akihiro.kusumi@oist.jp